# Robust Counterfactual Explanations on Graph Neural Networks

**Mohit Bajaj**[1*]   **Lingyang Chu**[2*]   **Zi Yu Xue**[1,3]   **Jian Pei**[4]
**Lanjun Wang**[1]   **Peter Cho-Ho Lam**[1]   **Yong Zhang**[1]

[1]Huawei Technologies Canada Co., Ltd.   [2]McMaster University
[3] The University of British Columbia   [4]Simon Fraser University
{mohit.bajaj1, zi.yu.xue, lanjun.wang, cho.ho.lam, yong.zhang3}@huawei.com
chul9@mcmaster.ca, jpei@cs.sfu.ca

## Abstract

Massive deployment of Graph Neural Networks (GNNs) in high-stake applications generates a strong demand for explanations that are robust to noise and align well with human intuition. Most existing methods generate explanations by identifying a subgraph of an input graph that has a strong correlation with the prediction. These explanations are not robust to noise because independently optimizing the correlation for a single input can easily overfit noise. Moreover, they are not counterfactual because removing an identified subgraph from an input graph does not necessarily change the prediction result. In this paper, we propose a novel method to generate robust counterfactual explanations on GNNs by explicitly modelling the common decision logic of GNNs on similar input graphs. Our explanations are naturally robust to noise because they are produced from the common decision boundaries of a GNN that govern the predictions of many similar input graphs. The explanations are also counterfactual because removing the set of edges identified by an explanation from the input graph changes the prediction significantly. Exhaustive experiments on many public datasets demonstrate the superior performance of our method.

## 1   Introduction

Graph Neural Networks (GNNs) [22, 37, 50] have achieved great practical successes in many real-world applications, such as chemistry [31], molecular biology [17], social networks [3] and epidemic modelling [34]. For most of these applications, explaining predictions made by a GNN model is crucial for establishing trust with end-users, identifying the cause of a prediction, and even discovering potential deficiencies of a GNN model before massive deployment. Ideally, an explanation should be able to answer questions like *"Would the prediction of the GNN model change if a certain part of an input molecule is removed?"* in the context of predicting whether an artificial molecule is active for a certain type of proteins [19, 41], *"Would an item recommended still be recommended if a customer had not purchased some other items in the past?"* for a GNN built for recommendation systems [9, 44].

Counterfactual explanations [28] in the form of *"If X had not occurred, Y would not have occurred"* [26] are the principled way to answer such questions and thus are highly desirable for GNNs. In the context of GNNs, a counterfactual explanation identifies a small subset of edges of the input graph instance such that removing those edges significantly changes the prediction made by the GNN. Counterfactual explanations are usually concise and easy to understand [28, 36] because they align well with the human intuition to describe a causal situation [26]. To make explanations more trustworthy, the counterfactual explanation should be robust to noise, that is, some slight changes on

---

*Equal contribution.

an input graph do not change the explanation significantly. This idea aligns well with the notion of robustness discussed for DNN explanations in computer vision domain [11]. According to Ghorbani et al. [11] many interpretations on neural networks are fragile as it is easier to generate adversarial perturbations that produce perceptively indistinguishable inputs that are assigned the same predicted label, yet have very different interpretations. Here, the concepts of "fragile" "robustness" describe the same concept from opposite perspectives. An interpretation is said to be fragile if systematic perturbations can lead to dramatically different interpretations without changing the label. Otherwise, the interpretation is said to be robust.

How to produce robust counterfactual explanations on predictions made by general graph neural networks is a novel problem that has not been systematically studied before. As to be discussed in Section 2, most GNN explanation methods [45, 25, 46, 37, 32] are neither counterfactual nor robust. These methods mostly focus on identifying a subgraph of an input graph that achieves a high correlation with the prediction result. Such explanations are usually not counterfactual because, due to the high non-convexity of GNNs, removing a subgraph that achieves a high correlation does not necessarily change the prediction result. Moreover, many existing methods [45, 25, 37, 32] are not robust to noise and may change significantly upon slight modifications on input graphs, because the explanation of every single input graph prediction is independently optimized to maximize the correlation with the prediction, thus an explanation can easily overfit the noise in the data.

In this paper[2], we develop RCExplainer, a novel method to produce robust counterfactual explanations on GNNs. The key idea is to first model the common decision logic of a GNN by set of decision regions where each decision region governs the predictions on a large number of graphs, and then extract robust counterfactual explanations by a deep neural network that explores the decision logic carried by the linear decision boundaries of the decision regions. We make the following contributions.

First, we model the decision logic of a GNN by a set of decision regions, where each decision region is induced by a set of linear decision boundaries of the GNN. We propose an unsupervised method to find decision regions for each class such that each decision region governs the prediction of multiple graph samples predicted to be the same class. The linear decision boundaries of the decision region capture the common decision logic on all the graph instances inside the decision region, thus do not easily overfit the noise of an individual graph instance. By exploring the common decision logic encoded in the linear boundaries, we are able to produce counterfactual explanations that are inherently robust to noise.

Second, based on the linear boundaries of the decision region, we propose a novel loss function to train a neural network that produces a robust counterfactual explanation as a small subset of edges of an input graph. The loss function is designed to directly optimize the explainability and counterfactual property of the subset of edges, such that: 1) the subgraph induced by the edges lies within the decision region, thus has a prediction consistent with the input graph; and 2) deleting the subset of edges from the input graph produces a remainder subgraph that lies outside the decision region, thus the prediction on the remainder subgraph changes significantly.

Last, we conduct comprehensive experimental study to compare our method with the state-of-the-art methods on fidelity, robustness, accuracy and efficiency. All the results solidly demonstrate the superior performance of our approach.

## 2 Related work

The existing GNN explanation methods [46, 37, 45, 32, 25] generally fall into two categories: model level explanation [46] and instance level explanation [37, 45, 32, 25].

A model level explanation method [46] produces a high-level explanation about the general behaviors of a GNN independent from input examples. This may be achieved by synthesizing a set of artificial graph instances such that each artificial graph instance maximizes the prediction score on a certain class. The weakness of model level explanation methods is that an input graph instance may not contain an artificial graph instance, and removing an artificial graph from an input graph does not necessarily change the prediction. As a result, model level explanations are substantially different from counterfactual explanations, because the synthesized artificial graphs do not provide insights into how the GNN makes its prediction on a specific input graph instance.

The instance level explanation methods [37, 45, 32, 25] explain the prediction(s) made by a GNN on a specific input graph instance or multiple instances by identifying a subgraph of an input graph

---

[2]Other versions of the paper are available at https://arxiv.org/abs/2107.04086

instance that achieves a high correlation with the prediction on the input graph. GNNExplainer [45] removes redundant edges from an input graph instance to produce an explanation that maximizes the mutual information between the distribution of subgraphs of the input graph and the GNN's prediction. Following the same idea by Ying et al. [45], PGExplainer [25] parameterizes the generation process of explanations by a deep neural network, and trains it to maximize a similar mutual information based loss used by GNNExplainer [45]. The trained deep neural network is then applied to generate explanations for a single input graph instance or a group of input graphs. MEG [30] incorporates strong domain knowledge in chemistry with a reinforcement learning framework to produce counterfactual explanations on GNNs specifically built for compound prediction, but the heavy reliance on domain knowledge largely limits its applicability on general GNNs. The recently proposed CF-GNNExplainer [24] independently optimizes the counterfactual property for each explanation but ignores the correlation between the prediction and the explanation.

Some studies [32, 37] also adapt the existing explanation methods of image-oriented deep neural networks to produce instance level explanations for GNNs. Pope et al. [32] extend several gradient based methods [33, 35, 49] to explain predictions made by GNNs. The explanations are prone to gradient saturation [12] and may also be misleading [1] due to the heavy reliance on noisy gradients. Velickovic et al. [37] extend the attention mechanism [7, 8] to identify the nodes in an input graph that contribute the most to the prediction. This method has to retrain the GNN with the altered architecture and the inserted attention layers. Thus, the explanations may not be faithful to the original GNN.

Instance level explanations from most of the methods are usually not counterfactual because, due to the non-convexity of GNNs, removing an explanation subgraph from the input graph does not necessarily change the prediction result. Moreover, those methods [45, 25, 37, 32, 24] are usually not robust to noise because the explanation of every single input graph prediction is independently optimized. Thus, an explanation can easily overfit the noise inside input graphs and may change significantly upon slight modifications on input graphs.

To tackle the weaknesses in the existing methods, in this paper, we directly optimize the counterfactual property of an explanation along with the correlation between the explanation and the prediction. Our explanations are also much more robust to modifications on input graphs, because they are produced from the common decision logic on a large group of similar input graphs, which do not easily overfit the noise of an individual graph sample.

Please note that our study is substantially different from adversarial attacks on GNNs. The adversarial attacking methods [51, 53, 42, 43, 20] use adversarial examples to change the predictions of GNNs but ignore the explainability of the generated adversarial examples [10]. Thus, the adversarial examples generated by adversarial attacks may not explain the original prediction.

Our method is substantially different from the above works because we focus on explaining the prediction by directly optimizing the counterfactual property of an explanation along with correlation of the explanation with the prediction. We also require that the explanation is generally valid for a large set of similar graph instances by extracting it from the common linear decision boundaries of a large decision region.

## 3   Problem Formulation

Denote by $G = \{V, E\}$ a graph where $V = \{v_1, v_2, \ldots, v_n\}$ is the set of $n$ nodes and $E \subseteq V \times V$ is the set of edges. The edge structure of a graph $G$ is described by an adjacency matrix $\mathbf{A} \in \{0, 1\}^{n \times n}$, where $\mathbf{A}_{ij} = 1$ if there is an edge between node $v_i$ and $v_j$; and $\mathbf{A}_{ij} = 0$ otherwise.

Denote by $\phi$ a GNN model that maps a graph to a probability distribution over a set of classes denoted by $C$. Let $D$ denote the set of graphs that are used to train the GNN model $\phi$. We focus on GNNs that adopt piecewise linear activation functions, such as MaxOut [14] and the family of ReLU [13, 15, 29].

The robust counterfactual explanation problem is defined as follows.

**Definition 1 (Robust Counterfactual Explanation Problem)** *Given a GNN model $\phi$ trained on a set of graphs $D$, for an input graph $G = \{V, E\}$, our goal is to explain why $G$ is predicted by the GNN model as $\phi(G)$ by identifying a small subset of edges $S \subseteq E$, such that (1) removing the set of edges in $S$ from $G$ that causes the maximum drop in the confidence of the original prediction; and (2) $S$ is stable and doesn't change when the edges and the feature representations of the nodes of $G$ are perturbed by random noise.*

In the definition, the first requirement requires that the explanation $S$ is counterfactual, and the second requirement requires that the explanation is robust to noisy changes on the edges and nodes of $G$.

## 4 Method

In this section, we first introduce how to extract the common decision logic of a GNN on a large set of graphs with the same predicted class. This is achieved by a decision region induced by a set of linear decision boundaries of the GNN. Then, based on the linear boundaries of the decision region, we propose a novel loss function to train a neural network that produces robust counterfactual explanations. Last, we discuss the time complexity of our method when generating explanations.

### 4.1 Modelling Decision Regions

Following the routines of many deep neural network explanation methods [33, 48], we extract the decision region of a GNN in the $d$-dimensional output space $\mathbb{O}^d$ of the last convolution layer of the GNN. The features generated by the last convolution layer are more conceptually meaningful and more robust to noise than those raw features of input graphs, such as vertices and edges [52, 2]. Denote by $\phi_{gc}$ the mapping function realized by the graph convolution layers that maps an input graph $G$ to its graph embedding $\phi_{gc}(G) \in \mathbb{O}^d$, and by $\phi_{fc}$ the mapping function realized by the fully connected layers that maps the graph embedding $\phi_{gc}(G)$ to a predicted distribution over the classes in $C$. The overall prediction $\phi(G)$ made by the GNN can be written as $\phi(G) = \phi_{fc}(\phi_{gc}(G))$.

For the GNNs that adopt piecewise linear activation functions for the hidden neurons, such as MaxOut [14] and the family of ReLU [13, 15, 29], the decision logic of $\phi_{fc}$ in the space $\mathbb{O}^d$ is characterized by a piecewise linear decision boundary formed by connected pieces of decision hyperplanes in $\mathbb{O}^d$ [1]. We call these hyperplanes **linear decision boundaries (LDBs)**, and denote by $\mathcal{H}$ the set of LDBs induced by $\phi_{fc}$. The set of LDBs in $\mathcal{H}$ partitions the space $\mathbb{O}^d$ into a large number of convex polytopes. A convex polytope is formed by a subset of LDBs in $\mathcal{H}$. All the graphs whose graph embeddings are contained in the same convex polytope are predicted as the same class [4]. Therefore, the LDBs of a convex polytope encode the common decision logic of $\phi_{fc}$ on all the graphs whose graph embeddings lie within the convex polytope [4]. Here, a graph $G$ is **covered** by a convex polytope if the graph embedding $\phi_{gc}(G)$ is contained in the convex polytope.

Based on the above insight, we model the **decision region** for a set of graph instances as a convex polytope that satisfies the following two properties. First, the decision region should be induced by a subset of the LDBs in $\mathcal{H}$. In this way, when we extract counterfactual explanations from the LDBs, the explanations are loyal to the real decision logic of the GNN. Second, the decision region should cover many graph instances in the training dataset $D$, and all the covered graphs should be predicted as the same class. In this way, the LDBs of the decision region capture the common decision logic on all the graphs covered by the decision region. Here, the requirement of covering a larger number of graphs ensures that the common decision logic is general, and thus it is less likely to overfit the noise of an individual graph instance. As a result, the counterfactual explanations extracted from the LDBs of the decision region are insensitive to slight changes in the input graphs. Our method can be easily generalized to incorporate prediction confidence in the coverage measure, such as considering the count of graphs weighted by prediction confidence. To keep our discussion simple, we do not pursue this detail further in the paper.

Next, we illustrate how to extract a decision region satisfying the above two requirements. The key idea is to find a convex polytope covering a large set of graph instances in $D$ that are predicted as the same class $c \in C$.

Denote by $D_c \subseteq D$ the set of graphs in $D$ predicted as a class $c \in C$, by $\mathcal{P} \subseteq \mathcal{H}$ a set of LDBs that partition the space $\mathbb{O}^d$ into a set of convex polytopes, and by $r(\mathcal{P}, c)$ the convex polytope induced by $\mathcal{P}$ that covers the largest number of graphs in $D_c$. Denote by $g(\mathcal{P}, c)$ the number of graphs in $D_c$ covered by $r(\mathcal{P}, c)$, and by $h(\mathcal{P}, c)$ the number of graphs in $D$ that are covered by $r(\mathcal{P}, c)$ but are not predicted as class $c$. We extract a decision region covering a large set of graph instances in $D_c$ by solving the following constrained optimization problem.

$$\max_{\mathcal{P} \subseteq \mathcal{H}} g(\mathcal{P}, c), \text{ s.t. } h(\mathcal{P}, c) = 0 \tag{1}$$

This formulation realizes the two properties of decision regions because $\mathcal{P} \subseteq \mathcal{H}$ ensures that the decision region is induced by a subset of LDBs in $\mathcal{H}$, maximizing $g(\mathcal{P}, c)$ requires that $r(\mathcal{P}, c)$ covers

a large number of graphs in $D_c$, and the constraint $h(\mathcal{P}, c) = 0$ ensures that all the graphs covered by $r(\mathcal{P}, c)$ are predicted as the same class $c$.

Once we find a solution $\mathcal{P}$ to the above problem, the decision region $r(\mathcal{P}, c)$ can be easily obtained by first counting the number of graphs in $D_c$ covered by each convex polytope induced by $\mathcal{P}$, and then select the convex polytope that covers the largest number of graphs in $D_c$.

## 4.2 Extracting Decision Regions

The optimization problem in Equation (1) is intractable for standard GNNs, mainly because it is impractical to compute $\mathcal{H}$, all the LDBs of a GNN. The number of LDBs in $\mathcal{H}$ of a GNN is exponential with respect to the number of neurons in the worst case [27]. To address this challenge, we substitute $\mathcal{H}$ by a sample $\tilde{\mathcal{H}}$ of LDBs from $\tilde{\mathcal{H}}$.

A LDB in the space $\mathbb{O}^d$ can be written as $\mathbf{w}^\top \mathbf{x} + b = 0$, where is $\mathbf{x} \in \mathbb{O}^d$ is a variable, $\mathbf{w}$ is the basis term, and $b$ corresponds to the bias. Following [4], for any input graph $G$, a linear boundary can be sampled from $\mathcal{H}$ by computing

$$\mathbf{w} = \frac{\partial \left( \max_1(\phi_{fc}(\boldsymbol{\alpha})) - \max_2(\phi_{fc}(\boldsymbol{\alpha})) \right)}{\partial \boldsymbol{\alpha}} |_{\boldsymbol{\alpha} = \phi_{gc}(G)}, \tag{2}$$

and

$$b = \max_1(\phi_{fc}(\boldsymbol{\alpha})) - \max_2(\phi_{fc}(\boldsymbol{\alpha})) - \mathbf{w}^T \boldsymbol{\alpha} |_{\boldsymbol{\alpha} = \phi_{gc}(G)}, \tag{3}$$

where $\max_1(\phi_{fc}(\boldsymbol{\alpha})))$ and $\max_2(\phi_{fc}(\boldsymbol{\alpha}))$ are the largest and the second largest values in the vector $\phi_{fc}(\boldsymbol{\alpha})$, respectively. Given an input graph $G$, Equations (2) and (3) identify one LDB from $\mathcal{H}$. Thus, we can sample a subset of input graphs uniformly from $D$, and use Equations (2) and (3) to derive a sample of LDBs as $\tilde{\mathcal{H}} \subset \mathcal{H}$.

Now, we substitute $\mathcal{H}$ in Equation (1) by $\tilde{\mathcal{H}}$ to produce the following problem.

$$\max_{\mathcal{P} \subseteq \tilde{\mathcal{H}}} g(\mathcal{P}, c), \text{ s.t. } h(\mathcal{P}, c) \leq \delta, \tag{4}$$

where $\delta \geq 0$ is a tolerance parameter to keep this problem feasible. The parameter $\delta$ is required because substituting $\mathcal{H}$ by $\tilde{\mathcal{H}}$ ignores the LDBs in $\mathcal{H} \setminus \tilde{\mathcal{H}}$. Thus, the convex polytope $r(\mathcal{P}, c)$ induced by subset of boundaries in $\tilde{\mathcal{H}}$ may contain instances that are not predicted as class $c$. We directly set $\delta = h(\tilde{\mathcal{H}}, c)$, which is the smallest value of $\delta$ that keeps the practical problem feasible.

The problem in Equation (4) can be proven to be a Submodular Cost Submodular Cover (SCSC) problem [18] (see Appendix D for proof) that is well known to be NP-hard [5]. We adopt a greedy boundary selection method to find a good solution to this problem [40]. Specifically, we initialize $\mathcal{P}$ as an empty set, and then iteratively select a new boundary $h$ from $\tilde{\mathcal{H}}$ by

$$h = \arg\min_{h \in \tilde{\mathcal{H}} \setminus \mathcal{P}} \frac{g(\mathcal{P}, c) - g(\mathcal{P} \cup \{h\}, c) + \epsilon}{h(\mathcal{P}, c) - h(\mathcal{P} \cup \{h\}, c)}, \tag{5}$$

where $g(\mathcal{P}, c) - g(\mathcal{P} \cup \{h\}, c)$ is the decrease of $g(\mathcal{P}, c)$ when adding $h$ into $\mathcal{P}$, and $h(\mathcal{P}, c) - h(\mathcal{P} \cup \{h\}, c)$ is the decrease of $h(\mathcal{P}, c)$ when adding $h$ into $\mathcal{P}$. Both $g(\mathcal{P}, c)$ and $h(\mathcal{P}, c)$ are non-increasing when adding $h \in \tilde{\mathcal{H}}$ into $\mathcal{P}$ because adding a new boundary $h$ may only exclude some graphs from the convex polytope $r(\mathcal{P}, c)$.

Intuitively, in each iteration, Equation (5) selects a boundary $h \in \tilde{\mathcal{H}}$ such that adding $h$ into $\mathcal{P}$ reduces $g(\mathcal{P}, c)$ the least and reduces $h(\mathcal{P}, c)$ the most. In this way, we can quickly reduce $h(\mathcal{P}, c)$ to be smaller than $\delta$ without decreasing $g(\mathcal{P}, c)$ too much, which produces a good feasible solution to the practical problem. We add a small constant $\epsilon$ to the numerator such that, when there are multiple candidates of $h$ that do not decrease $g(\mathcal{P}, c)$, we can still select the $h$ that reduces $h(\mathcal{P}, c)$ the most.

We apply a peeling-off strategy to iteratively extract multiple decision regions. For each class $c \in C$, we first solve the practical problem once to find a decision region $r(\mathcal{P}, c)$, then we remove the graphs covered by $r(\mathcal{P}, c)$ from $D_c$. If there are remaining graphs predicted as the class $c$, we continue finding the decision regions using the remaining graphs until all the graphs in $D_c$ are removed. When all the graphs in $D_c$ are removed for each class $c \in C$, we stop the iteration and return the set of decision regions we found.

### 4.3 Producing Explanations

In this section, we introduce how to use the LDBs of decision regions to train a neural network that produces a robust counterfactual explanation as a small subset of edges of an input graph. We form explanations as a subset of edges because GNNs make decisions by aggregating messages passed on edges. Using edges instead of vertices as explanations can provide better insights on the decision logic of GNNs.

#### 4.3.1 The Neural Network Model

Denote by $f_\theta$ the neural network to generate a subset of edges of an input graph $G$ as the robust counterfactual explanation on the prediction $\phi(G)$. $\theta$ represents the set of parameters of the neural network. For experiments, our explanation network $f$ consists of 2 fully connected layers with a ReLU activation and the hidden dimension of 64.

For any two connected vertices $v_i$ and $v_j$ of $G$, denote by $\mathbf{z}_i$ and $\mathbf{z}_j$ the embeddings produced by the last convolution layer of the GNN for the two vertices, respectively. The neural network $f_\theta$ takes $\mathbf{z}_i$ and $\mathbf{z}_j$ as the input and outputs the probability for the edge between $v_i$ and $v_j$ to be part of the explanation. This can be written as

$$\mathbf{M}_{ij} = f_\theta(\mathbf{z}_i, \mathbf{z}_j), \tag{6}$$

where $\mathbf{M}_{ij}$ denotes the probability that the edge between $v_i$ and $v_j$ is contained in the explanation. When there is no edge between $v_i$ and $v_j$, that is, $\mathbf{A}_{ij} = 0$, we set $\mathbf{M}_{ij} = 0$.

For an input graph $G = \{V, E\}$ with $n$ vertices and a trained neural network $f_\theta$, $\mathbf{M}$ is an $n$-by-$n$ matrix that carries the complete information to generate a robust counterfactual explanation as a subset of edges, denoted by $S \subseteq E$. Concretely, we obtain $S$ by selecting all the edges in $E$ whose corresponding entries in $\mathbf{M}$ are larger than 0.5.

#### 4.3.2 Training Model $f_\theta$

For an input graph $G = (V, E)$, denote by $S \subseteq E$ the subset of edges produced by $f_\theta$ to explain the prediction $\phi(G)$, our goal is to train a good model $f_\theta$ such that the prediction on the subgraph $G_S$ induced by $S$ from $G$ is consistent with $\phi(G)$; and deleting the edges in $S$ from $G$ produces a remainder subgraph $G_{E \setminus S}$ such that the prediction on $G_{E \setminus S}$ changes significantly from $\phi(G)$.

Since producing $S$ by $f_\theta$ is a discrete operation that is hard to incorporate in an end-to-end training process, we define two proxy graphs to approximate $G_S$ and $G_{E \setminus S}$, respectively, such that the proxy graphs are determined by $\theta$ through continuous functions that can be smoothly incorporated into an end-to-end training process.

The proxy graph of $G_S$, denoted by $G_\theta$, is defined by regarding $\mathbf{M}$ instead of $\mathbf{A}$ as the adjacency matrix. That is, $G_\theta$ has exactly the same graph structure as $G$, but the edge weights of $G_\theta$ is given by the entries in $\mathbf{M}$ instead of $\mathbf{A}$. Here, the subscript $\theta$ means $G_\theta$ is determined by $\theta$.

The proxy graph of $G_{E \setminus S}$, denoted by $G'_\theta$, also have the same graph structure as $G$, but the edge weight between each pair of vertices $v_i$ and $v_j$ is defined as

$$\mathbf{M}'_{ij} = \begin{cases} 1 - \mathbf{M}_{ij} & \text{if } \mathbf{A}_{ij} = 1 \\ 0 & \text{if } \mathbf{A}_{ij} = 0 \end{cases} \tag{7}$$

The edge weights of both $G_\theta$ and $G'_\theta$ are determined by $\theta$ through continuous functions, thus we can smoothly incorporate $G_\theta$ and $G'_\theta$ into an end-to-end training framework.

As discussed later in this section, we use a regularization term to force the value of each entry in $\mathbf{M}_{ij}$ to be close to either 0 or 1, such that $G_\theta$ and $G'_\theta$ better approximate $G_S$ and $G_{E \setminus S}$ respectively.

We formulate our loss function as

$$\mathcal{L}(\theta) = \sum_{G \in D} \left\{ \lambda \mathcal{L}_{same}(\theta, G) + (1 - \lambda)\mathcal{L}_{opp}(\theta, G) + \beta \mathcal{R}_{sparse}(\theta, G) + \mu \mathcal{R}_{discrete}(\theta, G) \right\}, \tag{8}$$

where $\lambda \in [0, 1]$, $\beta \geq 0$ and $\mu \geq 0$ are the hyperparameters controlling the importance of each term. The influence of these parameters is discussed in Appendix G. The first term of our loss function requires that the prediction of the GNN on $G_\theta$ is consistent with the prediction on $G$. Intuitively, this

means that the edges with larger weights in $G_\theta$ dominate the prediction on $G$. We formulate this term by requiring $G_\theta$ to be covered by the same decision region covering $G$.

Denote by $\mathcal{H}_G$ the set of LDBs that induce the decision region covering $G$, and by $|\mathcal{H}_G|$ the number of LDBs in $\mathcal{H}_G$. For the $i$-th LDB $h_i \in \mathcal{H}_G$, denote by $\mathcal{B}_i(\mathbf{x}) = \mathbf{w}_i^\top \mathbf{x} + b_i$, where $\mathbf{w}_i$ and $b_i$ are the basis and bias of $h_i$, respectively, and $\mathbf{x} \in \mathbb{O}^d$ is a point in the space $\mathbb{O}^d$. The sign of $\mathcal{B}_i(\mathbf{x})$ indicates whether a point $\mathbf{x}$ lies on the positive side or the negative side of $h_i$, and the absolute value $|\mathcal{B}_i(\mathbf{x})|$ is proportional to the distance of a point $\mathbf{x}$ from $h_i$. Denote by $\sigma(\cdot)$ the standard sigmoid function, we formulate the first term of our loss function as

$$\mathcal{L}_{same}(\theta, G) = \frac{1}{|\mathcal{H}_G|} \sum_{h_i \in \mathcal{H}_G} \sigma\left(-\mathcal{B}_i(\phi_{gc}(G)) * \mathcal{B}_i(\phi_{gc}(G_\theta))\right), \tag{9}$$

such that minimizing $\mathcal{L}_{same}(\theta, G)$ encourages the graph embeddings $\phi_{gc}(G)$ and $\phi_{gc}(G_\theta)$ to lie on the same side of every LDB in $\mathcal{H}_G$. Thus, $G_\theta$ is encouraged to be covered by the same decision region covering $G$.

The second term of our loss function optimizes the counterfactual property of the explanations by requiring the prediction on $G'_\theta$ to be significantly different from the prediction on $G$. Intuitively, this means that the set of edges with larger weights in $G_\theta$ are good counterfactual explanations because reducing the weights of these edges significantly changes the prediction. Following the above intuition, we formulate the second term as

$$\mathcal{L}_{opp}(\theta, G) = \min_{h_i \in \mathcal{H}_G} \sigma\left(\mathcal{B}_i(\phi_{gc}(G)) * \mathcal{B}_i(\phi_{gc}(G'_\theta))\right), \tag{10}$$

such that minimizing $\mathcal{L}_{opp}(\theta, G)$ encourages the graph embeddings $\phi_{gc}(G)$ and $\phi_{gc}(G'_\theta)$ to lie on the opposite sides of at least one LDB in $\mathcal{H}_G$. This further means that $G'_\theta$ is encouraged not to be covered by the decision region covering $G$, thus the prediction on $G'_\theta$ can be changed significantly from the prediction on $G$.

Similar to [45], we use a L1 regularization $\mathcal{R}_{sparse}(\theta, G) = \|\mathbf{M}\|_1$ on the matrix $\mathbf{M}$ produced by $f_\theta$ on an input graph $G$ to produce a sparse matrix $\mathbf{M}$, such that only a small number of edges in $G$ are selected as the counterfactual explanation. We also follow [45] to use an entropy regularization

$$\mathcal{R}_{discrete}(\theta, G) = -\frac{1}{|\mathbf{M}|} \sum_{i,j} (\mathbf{M}_{ij} \log(\mathbf{M}_{ij}) + (1 - \mathbf{M}_{ij}) \log(1 - \mathbf{M}_{ij})) \tag{11}$$

to push the value of each entry in $\mathbf{M}_{ij}$ to be close to either 0 or 1, such that $G_\theta$ and $G'_\theta$ approximate $G_S$ and $G_{E \setminus S}$ well, respectively.

Now we can use the graphs in $D$ and the extracted decision regions to train the neural network $f_\theta$ in an end-to-end manner by minimizing $\mathcal{L}(\theta)$ over $\theta$ using back propagation. Once we finish training $f_\theta$, we can first apply $f_\theta$ to produce the matrix $\mathbf{M}$ for an input graph $G = (V, E)$, and then obtain the explanation $S$ by selecting all the edges in $E$ whose corresponding entries in $\mathbf{M}$ are larger than 0.5. We do not need the extracted boundaries for inference as the the decision logic of GNN is already distilled into the explanation network $f$ during the training.

As discussed in Appendix B, our method can be easily extended to generate robust counterfactual explanations for node classification tasks.

Our method is highly efficient with a time complexity $O(|E|)$ for explaining the prediction on an input graph $G$, where $|E|$ is the total number of edges in $G$. Additionally, the neural network $f_\theta$ can be directly used without retraining to predict explanations on unseen graphs. Thus our method is significantly faster than the other methods [45, 32, 47, 38] that require retraining each time when generating explanations on a new input graph.

## 5 Experiments

We conduct series of experiments to compare our method with the state-of-the-art methods including GNNExplainer [45], PGExplainer [25], PGM-Explainer [38], SubgraphX [47] and CF-GNNExplainer [24]. For the methods that identify a set of vertices as an explanation, we use the set of vertices to induce a subgraph from the input graph, and then use the set of edges of the induced subgraph as the explanation. For the methods that identify a subgraph as an explanation, we directly use the set of edges of the identified subgraph as the explanation.

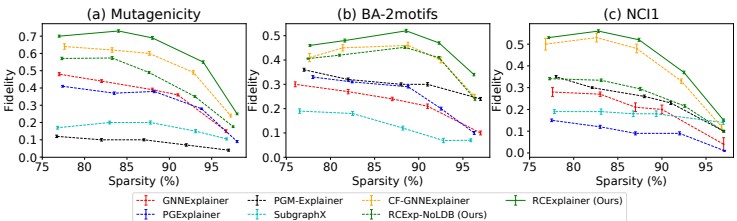

Figure 1: Fidelity performance averaged across 10 runs for the datasets at different levels of sparsity.

To demonstrate the effectiveness of the decision regions, we derive another baseline method named RCExp-NoLDB that adopts the general framework of RCExplainer but does not use the LDBs of decision regions to generate explanations. Instead, RCExp-NoLDB directly maximizes the prediction confidence on class $c$ for $G_\theta$ and minimizes the prediction confidence of class $c$ for $G'_\theta$.

We evaluate the explanation performance on two typical tasks: the graph classification task that uses a GNN to predict the class of an input graph, and the node classification task that uses a GNN to predict the class of a graph node. For the graph classification task, we use one synthetic dataset, BA-2motifs [25], and two real-world datasets, Mutagenicity [21] and NCI1 [39]. For the node classification task, we use the same four synthetic datasets as used by GNNExplainer [45], namely, BA-SHAPES, BA-COMMUNITY, TREE-CYCLES and TREE-GRID.

Limited by space, we only report here the key results on the graph classification task for fidelity, robustness and efficiency. Please refer to Appendix E for details on datasets, baselines and the experiment setups. Detailed experimental comparison on the node classification task will be discussed in Appendix F where we show that our method produces extremely accurate explanations. The code[3] is publicly available.

## 5.1 Fidelity

**Fidelity** is measured by the decrease of prediction confidence after removing the explanation (i.e., a set of edges) from the input graph [32]. We use fidelity to evaluate how counterfactual the generated explanations are on the datasets Mutagenicity, NCI1 and BA-2motifs. A large fidelity score indicates stronger counterfactual characteristics. It is important to note that fidelity may be sensitive to sparsity of explanations. The sparsity of an explanation $S$ with respect to an input graph $G = (V, E)$ is $sparsity(S, G) = 1 - \frac{|S|}{|E|}$, that is, the percentage of edges remaining after the explanation is removed from $G$. We only compare explanations with the same level of sparsity.

Figure 1 shows the results about fidelity. Our approach achieves the best fidelity performance at all levels of sparsity. The results validate the effectiveness of our method in producing highly counterfactual explanations. RCExplainer also significantly outperforms RCExp-NoLDB. This confirms that using LDBs of decision regions extracted from GNNs produces more faithful counterfactual explanations.

CF-GNNExplainer performs the best among the rest of the methods. This is expected as it optimizes the counterfactual behavior of the explanations which results in higher fidelity for the explanations in comparison to those produced by other methods such as GNNExplainer and PGExplainer.

The fidelity performance of SubgraphX reported in [47] was obtained by setting the features of nodes that are part of the explanation to $0$ but not removing the explanation edges from the input graph. This does not remove the message passing roles of the explanation nodes from the input graph because the edges connected to those nodes still can pass messages. In our experiments, we directly block the messages that are passed on the edges in the explanation, which completely prevents the explanation nodes in the input graph to participate in the message passing. As a result, the performance of SubgraphX drops significantly.

---

[3]Code available at `https://marketplace.huaweicloud.com/markets/aihub/notebook/detail/?id=e41f63d3-e346-4891-bf6a-40e64b4a3278`

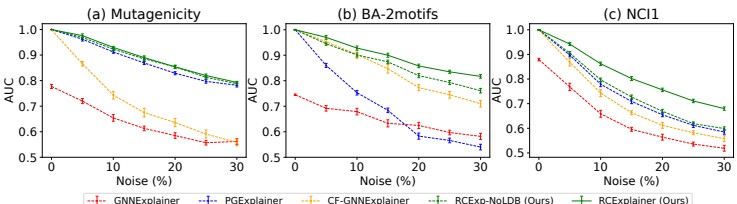

Figure 2: Noise robustness (AUC) averaged across 10 runs for the datasets at different levels of noise.

## 5.2 Robustness Performance

In this experiment, we evaluate the robustness of all methods by quantifying how much an explanation changes after adding noise to the input graph. For an input graph $G$ and the explanation $S$, we produce a perturbed graph $G'$ by adding random noise to the node features and randomly adding or deleting some edges of the input graph such that the prediction on $G'$ is consistent with the prediction on $G$. Using the same method we obtain the explanation $S'$ on $G'$. Considering top-$k$ edges of $S$ as the ground-truth and comparing $S'$ against them, we compute a receiver operating characteristic (ROC) curve and evaluate the robustness by the area under curve (AUC) of the ROC curve. We report results for $k = 8$ in Figure 2. Results for other values of $k$ are included in Appendix F where we observe similar trend.

Figure 2 shows the AUC of GNNExplainer, PGExplainer, RCExp-NoLDB and RCExplainer at different levels of noise. A higher AUC indicates better robustness. The percentage of noise shows the proportion of nodes and edges that are modified. Baselines such as PGM-Explainer and SubgraphX are not included in this experiment as they do not output the edge weights that are required for computing AUC. We present additional robustness experiments in Appendix F where we extend all the baselines to report node and edge level accuracy.

GNNExplainer performs the worst on most of the datasets, since it optimizes each graph independently without considering other graphs in the training set. Even when no noise is added, the AUC of GNNExplainer is significantly lower than 1 because different runs produce different explanations for the same graph prediction. PGExplainer is generally more robust than GNNExplainer because the neural network they trained to produce explanations implicitly considers all the graphs used for training. CF-GNNExplainer also performs worse than RCExplainer, which means it is more susceptible to the noise as compared to RCExplainer.

Our method achieves the best AUC on all the datasets, because the common decision logic carried by the decision regions of a GNN is highly robust to noise. PGExplainer achieves a comparable performance as our method on the Mutagenicity dataset, because the samples of this dataset share a lot of common structures such as carbon rings, which makes it easier for the neural network trained by PGExplainer to identify these structures in presence of noise. However, for BA-2motifs and NCI1, this is harder as samples share very few structures and thus the AUC of PGExplainer drops significantly. RCExplainer also significantly outperforms RCExp-NoLDB on these datasets which highlights the role of decision boundaries in making our method highly robust.

| Method | GNNExplainer | PGExplainer | PGM-Explainer | SubgraphX | CF-GNNExplainer | RCExplainer |
|--------|--------------|-------------|---------------|-----------|-----------------|-------------|
| **Time** | $1.2s \pm 0.2$ | **0.01s** $\pm 0.03$ | $13.1s \pm 3.9$ | $77.8s \pm 4.5$ | $4.6s \pm 0.2$ | **0.01s** $\pm 0.02$ |

Table 1: Average time cost for producing an explanation on a single graph sample.

**Efficiency.** We evaluate efficiency by comparing the average computation time taken for inference on unseen graph samples. Table 1 shows the results on the Mutagenicity dataset. Since our method also can be directly used for unseen data without any retraining, it is as efficient as PGExplainer and significantly faster than GNNExplainer, PGM-Explainer, SubgraphX and CF-GNNExplainer.

## 6 Conclusion

In this paper, we develop a novel method for producing counterfactual explanations on GNNs. We extract decision boundaries from the given GNN model to formulate an intuitive and effective

counterfactual loss function. We optimize this loss to train a neural network to produce explanations with strong counterfactual characteristics. Since the decision boundaries are shared by multiple samples of the same predicted class, explanations produced by our method are robust and do not overfit the noise. Our experiments on synthetic and real-life benchmark datasets strongly validate the efficacy of our method. In this work, we focus on GNNs that belong to Piecewise Linear Neural Networks (PLNNs). Extending our method to other families of GNNs and tasks such as link prediction, remains an interesting future direction.

Our method will benefit multiple fields where GNNs are intensively used. By allowing the users to interpret the predictions of complex GNNs better, it will promote transparency, trust and fairness in the society. However, there also exist some inherent risks. A generated explanation may expose private information if our method is not coupled with an adequate privacy protection technique. Also, some of the ideas presented in this paper may be adopted and extended to improve adversarial attacks. Without appropriate defense mechanisms, the misuse of such attacks poses a risk of disruption in the functionality of GNNs deployed in the real world. That said, we firmly believe that these risks can be mitigated through increased awareness and proactive measures.

## Acknowledgments and Disclosure of Funding

Lingyang Chu's research is supported in part by the startup grant provided by the Department of Computing and Software of McMaster University. All opinions, findings, conclusions, and recommendations in this paper are those of the authors and do not necessarily reflect the views of the funding agencies.

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
