# A    Illustration of RCExplainer's training

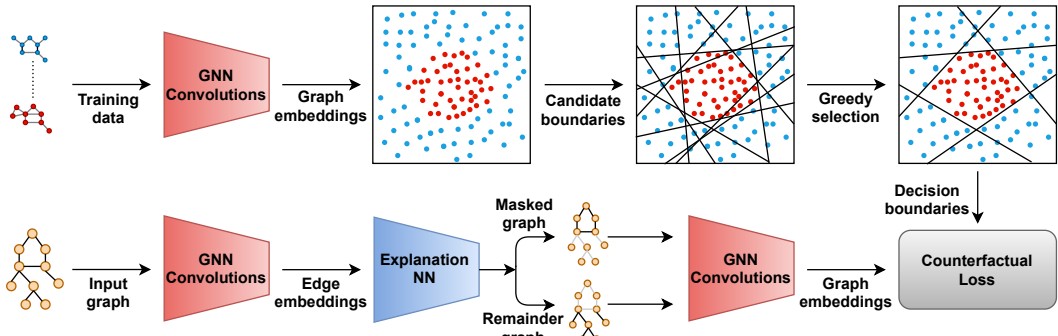

Figure 3: For training of RCExplainer, decision boundaries are extracted from the feature space of graph embeddings after the last graph convolution layer. After processing, a subset of boundaries is obtained and used to train an explanation neural network that takes edge activations from the convolution layers of GNN as input and predicts a mask over the adjacency matrix for the given graph sample. Counterfactual loss is used to optimize the explanation network.

# B    Node classification

Our method is directly applicable to the task of node classification with few simple modifications. Instead of extracting Linear Decision Boundaries (LDBs) in feature space of graph embeddings, we operate on the feature space of node embeddings obtained after the last graph convolution layer. We use the greedy method described in Equation (5) to find the decision regions for each class, except for the node classification, the functions $g(\cdot)$ and $h(\cdot)$ denote the coverage of nodes rather than graphs.

The next step to train the explanation network $f_\theta$ to generate counterfactual explanations for node classification is identical to the procedure described in Section 4 except for one difference. For node classification, since a node's prediction is only influenced by its local neighborhood, therefore we only need to consider the computation graph of the given node while generating the explanation. The computation graph of a node is defined as $k$-hop neighborhood of the node , where $k$ refers to number of graph convolution layers in the given GNN model $\phi$. In other words, GNN performs $k$ steps of message passing through its graph convolution layers during the forward pass to effectively convolve $k$-hop neighborhood of the given node. Hence, the output of $f_\theta$ is the output mask over the adjacency matrix of the computation graph of the given node. The edges with mask values more than 0.5 are chosen from the computation subgraph to form the explanation subset $S$ that can explain the original node classification prediction.

# C    Interpreting individual boundaries

We present a case study to demonstrate that our method can be adapted to answer the question, *"Which substructures make the samples of one class different from the samples of other specific class, and therefore can be masked to flip the prediction between the two given classes?"*. This is useful in various fields, for instance, in drug discovery where the classes correspond to different chemical properties possible of a drug compound, researchers are often interested in understanding the role of chemical structures that result in a prediction of a particular property instead of another specific one. Also, this is especially helpful for debugging in cases where one expects a particular output for the given input but the GNN's prediction does not agree with the expectation.

This case corresponds to a more constrained setting of counterfactual explanations as the target prediction is also predetermined. Let the original predicted label and the target label on a given graph $G$ be denoted by $c_i$ and $c_j$ respectively. Since our method explicitly models the boundaries separating the samples of one class from the samples of other classes, our method can be easily adapted to answer such questions. If we are able to only interpret the boundary separating the samples of the given two classes, this would allow us to uncover the substructures that make the samples of first

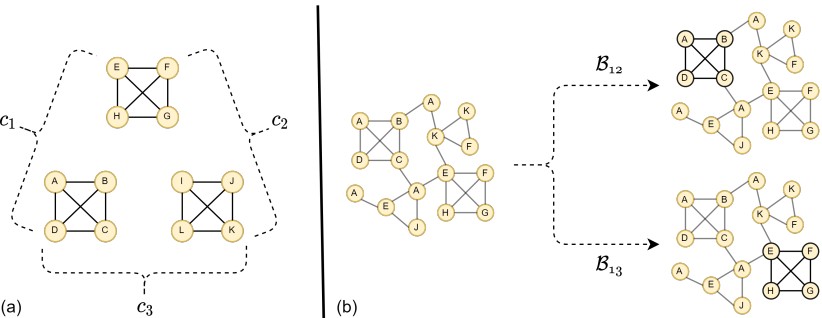

Figure 4: (a) Three classes and the motifs associated with each class are shown. All samples of the same class contain same two motifs. (b) Explanation results for an instance of class $c_1$ are shown w.r.t both of the boundaries separately. In both cases, RCExplainer correctly identifies the motif (highlighted in black) that is associated with the class $c_1$ but not associated with the class that lies on the other side of the given boundary.

class different from the samples of the other class. To address this, we modify the loss terms to

$$\mathcal{L}_{same}(\theta, G) = \sigma(-\mathcal{B}_{ij}(\phi_{gc}(G)) * \mathcal{B}_{ij}(\phi_{gc}(G_\theta))), \tag{12}$$

$$\mathcal{L}_{opp}(\theta, G) = \sigma(\mathcal{B}_{ij}(\phi_{gc}(G)) * \mathcal{B}_{ij}(\phi_{gc}(G'_\theta))), \tag{13}$$

where $\mathcal{B}_{ij}$ refers to the specific boundary in the set $\mathcal{P}$ separating the samples with predicted label $c_i$ from the samples with the predicted label $c_j$. Since we are only concerned about changing the outcome from $c_i$ to $c_j$, we need to consider only the boundary separating these classes while formulating the loss for the network.

We verify this on a synthetic graph classification dataset with 3 classes, $c_1$, $c_2$ and $c_3$ such that each graph sample contains exactly 2 motifs. Both the motifs jointly determine the class because each possible pair of classes share exactly one motif as shown in Figure 4(a). We show explanation results produced by RCExplainer on an instance of class $c_1$ in Figure 4(b). For a given graph sample of class $c_1$, we separately find explanations with respect to each of the two boundaries $\mathcal{B}_{12}$ and $\mathcal{B}_{13}$, $\mathcal{B}_{12}$ separates $c_1$ from $c_2$, while $\mathcal{B}_{13}$ separates $c_1$ from $c_3$. We can see in the Figure 4(b) that optimizing our method w.r.t $\mathcal{B}_{12}$ correctly identifies the motif (ABCD) in the sample that is not associated with the class $c_2$. The other motif (EFGH) which is also associated with the $c_2$ is not considered important by the method. When we find the explanation for the same graph sample but with respect to the boundary $\mathcal{B}_{13}$, the results are opposite and align with the expectations. In this case, the motif (EFGH) that is not associated with $c_3$ is highlighted instead of the motif (ABCD). We observe similar behavior on the instances of other classes where interpreting an instance with respect to a single boundary correctly identifies the motif that identifies the given class from the other class.

In conclusion, the above case study demonstrates that our method can highlight the motif unique to the class $c_i$ by interpreting the boundary $\mathcal{B}_{ij}$ separating the classes $c_i$ and $c_j$. Removing the highlighted motif from the given sample causes the drop in confidence of original predicted label $c_i$ while increasing the confidence for the class $c_j$.

## D  Proof: Decision region extraction is an instance of SCSC optimization

Now we prove that the optimization problem in Equation (4) is an instance of Submodular Cover Submodular Cost (SCSC) problem.

The Equation (4) can be written as

$$\min_{\mathcal{P} \subseteq \tilde{\mathcal{H}}} D_c - g(\mathcal{P}, c), \text{ s.t. } D'_c - h(\mathcal{P}, c) \geq D'_c - \delta. \tag{14}$$

Maximizing $g(\mathcal{P}, c)$ denotes maximizing the coverage of the set of boundaries $\mathcal{P}$ for the samples of class $c$ denoted by $D_c$. This can be seen as minimizing $D_c - g(\mathcal{P}, c)$ which denotes the number of

graph samples of class $c$ that are not covered by $g(\mathcal{P}, c)$ and thus exclusive to $D_c$. $D'_c$ in the constraint is equal to $D - D_c$ that denotes the set of graph samples in the dataset $D$ that do not belong to the class $c$.

Let us denote $D_c - g(\mathcal{P}, c)$ by function $g'(\mathcal{P})$ and $D'_c - h(\mathcal{P}, c)$ by $h'(\mathcal{P})$. To prove that the optimization problem in Equation (14) is an instance of SCSC problem, we prove the functions $g'(\mathcal{P})$ and $h'(\mathcal{P})$ are submodular with respect to $\mathcal{P}$.

For function $g'(\mathcal{P})$ to be submodular with respect to $\mathcal{P}$, we show that for any two arbitrary sets of LDBs denoted by $\mathcal{P} \subseteq \tilde{\mathcal{H}}$ and $\mathcal{Q} \subseteq \tilde{\mathcal{H}}$, if $\mathcal{P} \subseteq \mathcal{Q}$ then

$$g'(\mathcal{P} + \{h\}) - g'(\mathcal{P}) \geq g'(\mathcal{Q} + \{h\}) - g'(\mathcal{Q}) \tag{15}$$

is always satisfied for a linear decision boundary $h \in \tilde{\mathcal{H}} \setminus \mathcal{Q}$.

As discussed in Section 4 the LDBs in $\mathcal{P}$ induce a convex polytope $r(\mathcal{P}, c)$ that has the maximum coverage of samples of class $c$. Adding a new boundary $h$ to $\mathcal{P}$ may remove (separate) some samples of class $c$ from $r(\mathcal{P}, c)$ and lower its coverage. This reduction in coverage is denoted by the term $g'(\mathcal{P} + \{h\}) - g'(\mathcal{P})$ on the left hand side of Equation (15). Similarly the term $g'(\mathcal{Q} + \{h\}) - g'(\mathcal{Q})$ on the right hand side of Equation (15) denotes the reduction in coverage for the subset $\mathcal{Q}$.

Now, since $\mathcal{P} \subseteq \mathcal{Q}$, the set of graph samples contained in the polytope $r(\mathcal{Q}, c)$ is subset of the graph samples contained in the polytope $r(\mathcal{P}, c)$. Hence, adding new a LDB $h$ to $\mathcal{P}$ is not going to remove less number of samples from the polytope $r(\mathcal{P}, c)$ as compared to the samples removed from the polytope $r(\mathcal{Q}, c)$. Therefore, the function $g'(\mathcal{P})$ is submodular with respect to $\mathcal{P}$.

Similarly, we can prove the function $h'(\mathcal{P})$ to be submodular with respect to $\mathcal{P}$. This concludes the proof.

# E   Implementation details

**Datasets.**   Table 2 shows the properties of all the datasets used in experiments. The last row corresponds to the test accuracy of the GCN model we train on the corresponding dataset.

| | BA-SHAPES | BA-COMMUNITY | TREE-CYCLES | TREE-GRID | BA-2motifs | Mutagenicity | NCI1 |
|---|---|---|---|---|---|---|---|
| # of Nodes (avg) | 700 | 1400 | 871 | 1020 | 25 | 30.32 | 29.87 |
| # of Edges (avg) | 2050 | 4460 | 970 | 2540 | 25.48 | 30.77 | 32.30 |
| # of Graphs | 1 | 1 | 1 | 1 | 700 | 4337 | 4110 |
| # of Classes | 4 | 8 | 2 | 2 | 2 | 2 | 2 |
| Base | BA graph | BA graph | Tree | Tree | BA graph | — | — |
| Motifs | House | House | Cycle | Grid | House & Cycle | — | — |
| License | Apache 2.0 | Apache 2.0 | Apache 2.0 | Apache 2.0 | — | — | — |
| Test accuracy | 0.98 | 0.95 | 0.99 | 0.99 | 0.91 | 0.91 | 0.84 |

Table 2: Properties of the datasets used and the test accuracy of the corresponding trained GCN models.

**Baselines.**   For the baselines, we use publically available implementations provided in [45, 16, 38, 23] to obtain the results. Implementation of GNNExplainer provided by [45] is licensed under Apache 2.0 license while implementation of SubgraphX provided by [23] is licensed under GNU General Public License v3.0. We use the default parameters provided by the authors for the baselines.

For the local baseline RCExp-NoLDB, we use same setup as RCExplainer except we don't use LDBs for training the explanation network $f$. The loss function denoted by $\mathcal{L}_{conf}$ for this baseline aligns with the loss functions of GNNExplainer and PGExplainer except we introduce a second term to enforce the counterfactual characteristics. We directly maximize the confidence of the original predicted class $c$ on the masked graph $G_\theta$ and minimize the confidence of original predicted class for the remainder graph $G'_\theta$. $\mathcal{L}_{conf}$ can be expressed as :

$$\mathcal{L}_{conf}(\theta, G) = -\log(P_\phi(Y = c | X = G_\theta)) - \frac{\eta}{\log(P_\phi(Y = c | X = G'_\theta))}, \tag{16}$$

where $P_\phi(Y|X = G_x)$ corresponds to the conditional probability distribution learnt by GNN model $\phi$ for $G_x$ as input graph. $Y$ corresponds to the random variable representing the set of classes $\mathcal{C}$ and $X$ is the random variable representing possible input graphs for the GNN $\phi$. Here $\eta$ is a hyperparameter that represents the weight of the second term in the loss function. The loss $\mathcal{L}_{conf}$ is jointly minimized with the regularizers $\mathcal{R}_{sparse}$ and $\mathcal{R}_{discrete}$ specified in Section 4.

**Training details.** We follow [45, 25] and use the same architecture to train a GNN model with 3 graph convolution layers for generating explanations on each dataset. Consistent with prior works [45, 25, 38], we use (80/10/10)% random split for training/validation/test for each dataset.

We use Adam optimizer to tune the parameters of $f_\theta$ and set learning rate to $0.001$. We train our method for 600 epochs. For node classification datasets, we set $\lambda$ to $0.85$, $\beta$ to $0.006$ and $\mu$ to $0.66$. For graph classification datasets, we set $\lambda$ to $0.1$, $\mu$ to $0.66$. $\beta$ is set to $6 \times 10^{-5}$ for BA-2motifs and NCI1, and to $6 \times 10^{-4}$ for Mutagenicity. We also scale the combined loss by factor of $15$ for all the datasets. The number of LDBs to be sampled from GNN for each class is set to $50$. Empirically, we find that this is enough as the subset of LDBs selected greedily from this set is able to cover all the samples of the given class. Our codebase is built on the top of implementations provided by [45, 25].

All of the experiments are conducted on a Linux machine with an Intel i7-8700K processor and a Nvidia GeForce GTX 1080 Ti GPU with 11GB memory. Our code is implemented using python 3.8.5 with Pytorch 1.8.1 that uses CUDA version 10.0.

# F  Additional experiments

**Fidelity.** As described in Section 5, counterfactual characteristic of an explanation is measured by using fidelity as an evaluation metric. It is defined as drop in confidence of the original predicted class after masking the produced explanation in the original graph [32]. Since, we produce explanations as edges, we mask the edges in the input graph to calculate the drop. Fidelity for the input graph $G$ and the produced explanation $S$ is formally written as

$$fidelity(S, G) = P_\phi(Y = c|X = G) - P_\phi(Y = c|X = G_{E\setminus S}), \qquad (17)$$

where $c$ denotes the class predicted by $\phi$ for $G$. As discussed in Section 5, explanations are mostly useful, if they are sparse (concise). Sparsity is defined as the fraction of total edges that are present in $E$ but not in $S$:

$$sparsity(S, G) = 1 - \frac{|E|}{|S|}, \qquad (18)$$

However, since the approaches like SubgraphX and PGM-Explainer do not report the importance ranking of edges of $G$, it's not feasible to completely control the edge sparsity of the desired explanation. Hence, we take samples with similar sparsity level for comparison. Consistent with prior works [45, 25], we compute fidelity for the samples that are labelled positive, for instance in Mutagenicity dataset, we compute fidelity for the compounds that are labelled as mutagenic. The results are presented in Figure 1.

**Robustness to noise.**

As discussed in Section 5, we use AUC to compare robustness of different methods. AUC is defined as area under receiver operating characteristic (ROC) curve of a binary classifier. We consider the top-$k$ edges of the produced explanation $S$ for the input graph $G$ as ground-truth. After we obtain the explanation $S'$ for the noisy sample $G'$, we formulate this as binary classification problem. For each edge in $G'$, if it is present in the top-$k$ edges of the produced explanation $S$, then it is labeled positive, and negative otherwise. For $G'$, the mask weight of an edge predicted by explanation network $f_\theta$ is the probability of the corresponding edge being classified as positive. Limited by space, we only reported the results for $k = 8$ in Section 5. Now, we report the results for $k = 4$ and $k = 12$ in Figure 5 where we observe similar trend as observed in Figure 2. RCExplainer outperforms the rest of the methods by big margin on BA-2motifs and NCI1.

Since, AUC evaluation requires that the explanation method outputs the importance weights for the edges of a noisy sample $G'$, we cannot use this for comparing approaches like SubgraphX and PGM-Explainer that do not provide this information. Therefore, to provide a more comprehensive comparison, we use node accuracy as a measure to compare all the baselines. For calculating node accuracy, we consider top-$k$ important nodes in the explanation for the original graph $G$ as ground-truth and compare them with the the top-$k$ important nodes obtained through the explanation for the

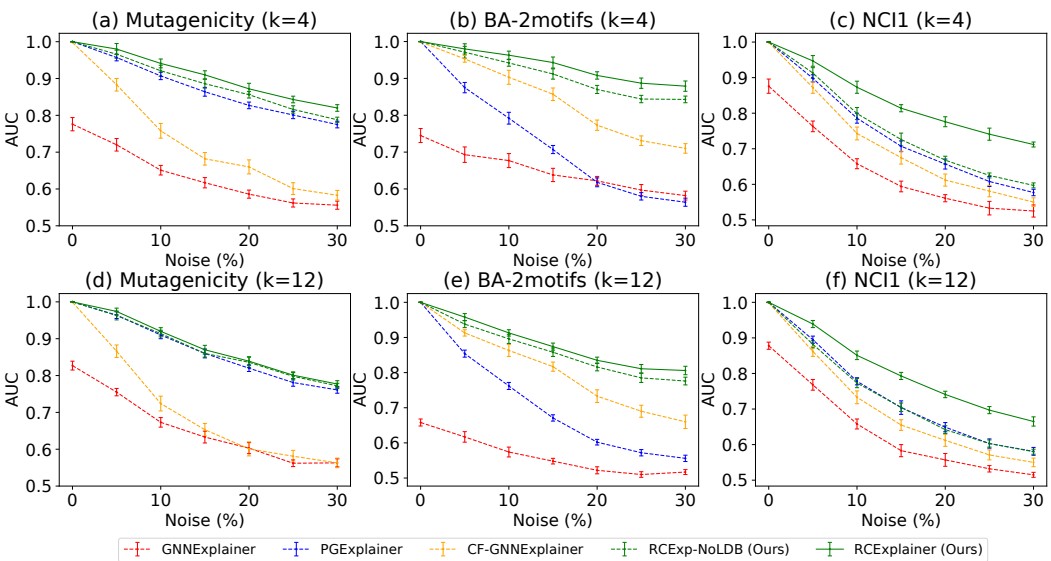

Figure 5: Noise robustness (AUC) averaged across 10 runs for the datasets at different levels of noise.

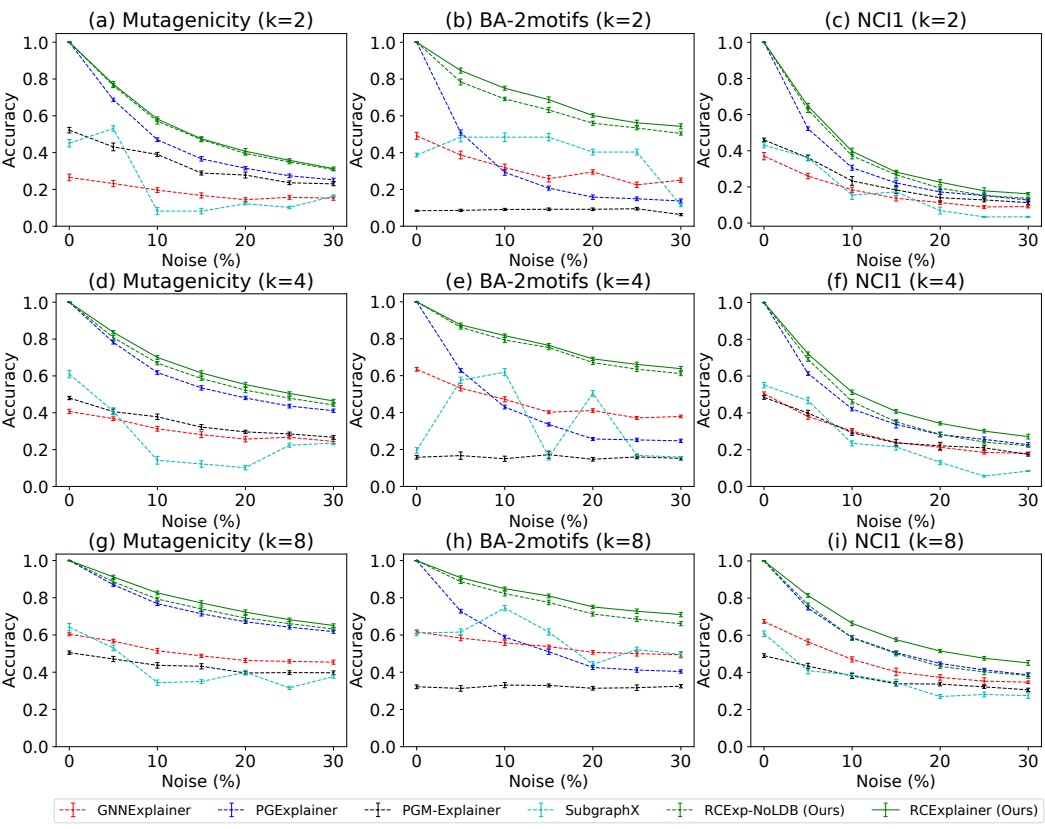

Figure 6: Noise robustness (node accuracy) averaged across 10 runs for the datasets at different levels of noise.

noisy graph $G'$. However, the challenge is that GNNExplainer, PGExplainer, RCExp-NoLDB and RCExplainer do not rank nodes based on their importance. To address this, we use edge weights to

obtain the node weights. We approximate the node weights as :

$$\mathbf{a}_i = \max_{j \in \{1,...,|V|\}} (\mathbf{M}_{ij}), \quad (19)$$

where $\mathbf{a}_i$ denotes the weight of the node $v_i$ and $\mathbf{M}$ is the weighted adjacency mask predicted by $f_\theta$. We believe this is a valid approximation because for an important edge to exist in the explanation subgraph, nodes connected by this edge must also be considered important and be present in the explanation subgraph. Now, using these node weights, we can obtain the ground-truth set of nodes by picking top-$k$ important nodes of the explanation on $G$. Comparing top-$k$ important nodes of explanation on $G'$ with the ground-truth set of nodes gives us accuracy.

We present the node accuracy plots for $k = 2, 4$ and $8$ in Figure 6. We also note that comparison is not completely fair to GNNExplainer, PGExplainer and our method because of the approximation used to extend these methods for computing node level accuracy. Despite the approximation, our method significantly outperforms all the methods. GNNExplainer, PGM-Explainer and SubgraphX do not perform very well because they optimize each sample independently to obtain an explanation.

**Node classification.**

| | BA-SHAPES | | BA- COMMUNITY | | TREE-CYCLES | | TREE-GRID | |
|---|---|---|---|---|---|---|---|---|
| | AUC | Acc | AUC | Acc | AUC | Acc | AUC | Acc |
| GNNExplainer | 0.925 | 0.729 | 0.836 | 0.750 | 0.948 | 0.862 | 0.875 | 0.741 |
| PGExplainer | 0.963 | 0.932 | 0.945 | 0.872 | 0.987 | 0.924 | 0.907 | 0.871 |
| PGM-Explainer | *n.a.* | 0.965 | *n.a.* | **0.926** | *n.a.* | 0.954 | *n.a.* | 0.885 |
| CF-GNNExplainer | *n.a.* | 0.960 | *n.a.* | *n.a.* | *n.a.* | 0.940 | *n.a.* | 0.960 |
| RCExplainer (Ours) | **0.998** | **0.973** | **0.995** | 0.916 | **0.993** | **0.993** | **0.995** | **0.974** |
| | ± 0.001 | ± 0.003 | ± 0.002 | ± 0.009 | ± 0.003 | ± 0.003 | ± 0.002 | ± 0.005 |

Table 3: AUC and accuracy evaluation on synthetic node classification datasets.

We evaluate our method on four synthetic node classification datasets used by GNNExplainer [45], namely, BA-SHAPES, BA-COMMUNITY, TREE-CYCLES and TREE-GRID. Following [45, 25], we formalize the explanation problem as binary classification of edges and adopt AUC under the ROC curve as the evaluation metric. This evaluation is only possible for synthetic datasets where we can consider the motifs as reasonable approximations of the explanation ground truth. The edges that are part of a motif are positive and rest of the edges are labelled negative during the evaluation. We show the results in Table 3 where we demonstrate that our method is extremely accurate and achieves close to optimal score for AUC on all of the datasets. This is solid evidence of our method's ability to capture the behavior of underlying GNN better and produce consistently accurate explanations to justify the original predictions. Please note that PGM-Explainer does not provide edge weights so it is not applicable for AUC. Also since the implementation of CF-GNNExplainer is not available, we only report those results that are available in [24].

# G    Hyperparameter analysis

**Number of LDBs.**

As mentioned in Section 4, we sample LDBs from the decision logic of GNN to form a candidate pool from which some boundaries are selected by the greedy method. In Figure 7, we show the effect of number of sampled candidate boundaries on the performance on BA-Community dataset. As we increase the number of sampled LDBs from 10 to 50, the fidelity improves and saturation is achieved once 50 LDBs are sampled. This is consistent with the expectations as more boundaries are sampled, the quality of decision region improves. When there are enough boundaries that can result in a good decision region after greedy selection, the performance saturates.

**Choosing $\lambda$.**

in Figure 8, we show the effect of $\lambda$ hyperparameter introduced in Equation (8). We show the fidelity performance of our method on BA-2motifs dataset for different values of $\lambda$ ranging from 0.01 to 0.90. The fidelity results are worst for $\lambda = 0.9$ as the second term $\mathcal{L}_{opp}$ of the loss that enforces counterfactual behavior of explanations is weighted very less in the combined loss. Setting $\lambda = 0.1$ gives the best results for fidelity.

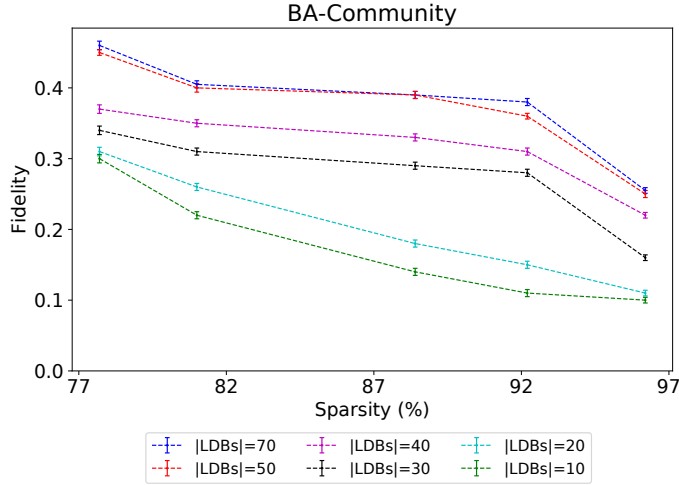

Figure 7: Fidelity vs Sparsity plots on BA-Community for different number of sampled LDBs.

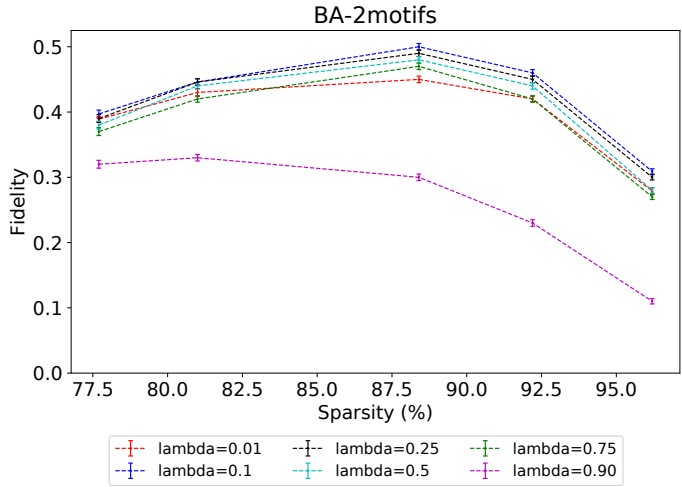

Figure 8: Fidelity vs Sparsity plots on BA-2motifs for different values of $\lambda$.

## H    Qualitative results

**Qualitative results.** We present the sample results produced by GNNExplainer, PGExplainer and RCExplainer in Table 4. Our method consistently identifies the right motifs with high precision and is also able to handle tricky cases. For instance in Figure (q), note that our method is able to identify the right motif in the presence of another "house-structure". The other structure contains the query node but as it also contains the nodes from the other community, hence, it is not the right explanation for the prediction on the given query node. In Figure (t), our method is able to correctly identify both NO2 groups present in the compound, and as discussed before, NO2 groups attached to carbon rings are known to make the compounds mutagenic [6]. The edges connecting nitrogen(N) atoms to the carbon(C) atoms are given the highest weights in the explanation. This is very intuitive in counterfactual sense as masking these edges would break the NO2 groups from the carbon ring and push the prediction of the compound towards "Non-Mutagenic".

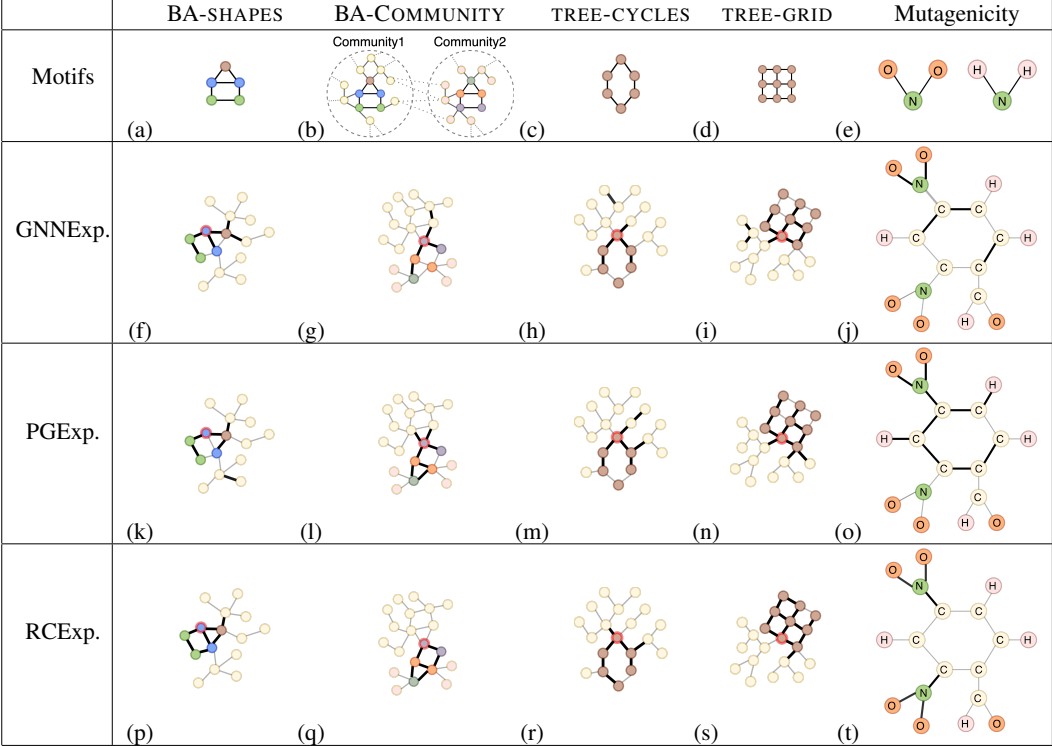

Table 4: Qualitative results produced by GNNExplainer, PGExplainer and RCExplainer. The motifs present in the corresponding dataset are shown in the first row and the corresponding explanations are shown in the later rows. First four columns correspond to node classification where the node highlighted in red is being explained and the node colors denote different labels. Last column corresponds to graph classification where the prediction is explained for a mutagenic sample and the colors denote different atoms. Explanations are highlighted in black.