# OpenReview forum: "Robust Counterfactual Explanations on Graph Neural Networks"
_NeurIPS.cc/2021/Conference — NeurIPS 2021 Poster_

### Official Review · Reviewer_BMo4 · 2021-07-08

**Rating:** 6
**Confidence:** 2

**Summary:**

This paper proposes a GNN explanation model to generate counterfactural and noise-robust explanations. The explanations are produced by a neural network as a set of edges where before and after removing them will result in a large deviation in the final prediction. To train such as neural network, instead of directly maximizing the deviation (in terms of confidence score) before and after removing the edges, the paper leverages common decision boundaries summarized from the linear decision boundaries of the GNN over the training graphs so that removing the edges cause many violations of the common decision boundaries. This way, the explanation is more robust to noise and does not overfit to individual instances. Experiments demonstrate the fidelity and robustness of the proposed method.

**Limitations And Societal Impact:**

Yes. The authors comment on the potential privacy concerns of the generated explanations.

**Main Review:**

Quality: Overall, I feel the paper makes a solid contribution for explaining GNNs. The motivation of generating conunterfactual explanations robust to noise is convincing, and the proposed solution is technically sound. By summarizing common decision boundaries from the linear decision boundaries of the GNN, the proposed method find edges removing which cause the maximum violations of these decision boundaries. It makes sense to me to use common decision boundaries instead of prediction scores to measure the "importance" of an edge, because it is more robust to noise and generates explanations based on consistency with the whole training data instead of individuals. The experiments on fidelity and robustness validate that the method generates high-quality and robust-to-noise explanations for GNNs. One concern is that many approximations are made to transform those NP-hard optimization problems to greedy/continuous problems. This results in no gurantees in the solutions. I would like the authors to explain more about the effects (on solution quality and efficiency) of such approximations.

Originality and Significance: Since I am not an expert in GNN explanability, I can hardly evaluate the originality and significance of the method, such as how counterfactual explanations are generated for models other than GNNs, and whether the proposed method makes a simple modification to existing methods for explaining general neural networks. I will listen to other reviewers' comments on this part.

Clarity: The writing of this paper can be improved. For example, "common decision boundary" appears first in the abstract, however its concrete explanation does not appear until section 4.1. I would suggest avoid such technical terms in abstract, and explain (define) "common decision boundary/logic" explicitly in the introduction. Also, the sentence "Our explanations are naturally robust to noise because they are produced from the common decision boundaries of a GNN that govern the predictions of many similar input graphs" is not straightforward to me. I spent a hard time trying to understand why the proposed method is robust to noise, until I understand common decision boundary in the very late part of the paper.

I give a score 6 with a low confidence due to my limited knowledge in this area. I am willing to adjust my score if other reviewers help endorse its originality and significance.

**Time Spent Reviewing:**

3

---

> ### Author Response · Authors · 2021-08-10
> **Response to Reviewer BMo4**
>
> We thank the reviewer for valuable comments. Below is our response.
>
> **Q1: One concern is that many approximations are made to transform those NP-hard optimization problems to greedy/continuous problems. This results in no guarantees in the solutions. I would like the authors to explain more about the effects (on solution quality and efficiency) of such approximations.**
>
> **A1**: Like most existing GNN explanation methods, it is difficult to obtain a theoretical guarantee on the quality of our explanations due to the complicated structure of GNNs. Therefore, we demonstrate the superior performance of our method by extensive experiments, as the existing studies in literature did.
>
> One technical contribution we made in this paper is to prove that the optimization problem in Equation (4) is a SCSC problem, which is NP-hard according to [5]. The SCSC problem has been well investigated in literature. Since solving the SCSC problem is not the major focus of this paper, we adopt a classical greedy solution proposed by [36]. [36] also establishes in-depth theoretical analysis on the complexity as well as the solution quality of the greedy algorithm. We will clarify this in the revised paper.
>
>
> **Q2: Originality and Significance: Since I am not an expert in GNN explainability, I can hardly evaluate the originality and significance of the method, such as how counterfactual explanations are generated for models other than GNNs, and whether the proposed method makes a simple modification to existing methods for explaining general neural networks. I will listen to other reviewers' comments on this part**
>
> **A2**: As discussed in lines 36-45 and in the Introduction section, generating robust counterfactual explanations on graph neural networks is a novel problem that has not been systematically studied before.
>
> Explanations produced by most existing methods are also not robust to random noise. This is because they independently optimize the correlation between a single input graph and the corresponding prediction made by a GNN, but do not explicitly leverage the real decision logic of GNNs that is shared by a group of input graphs. As a result, some noisy nodes and edges may achieve a very high correlation with the prediction score that might cause these methods to miss patterns that generally exist in many similar input graphs and are the right explanations of GNN’s predictions. Due to this sensitivity to noise, even small modifications when applied to an input graph that do not change the GNN’s prediction may cause the explanation to change dramatically.
>
> To the best of our knowledge, we are the first to leverage decision boundaries of a model to produce robust counterfactual explanations. No other attention mechanism applicable to general neural networks rely on the extracted decision boundaries to produce such explanations.
>
>
> **Q3: Clarity: The writing of this paper can be improved. For example, "common decision boundary" appears first in the abstract, however its concrete explanation does not appear until section 4.1. I would suggest avoid such technical terms in abstract, and explain (define) "common decision boundary/logic" explicitly in the introduction. Also, the sentence "Our explanations are naturally robust to noise because they are produced from the common decision boundaries of a GNN that govern the predictions of many similar input graphs" is not straightforward to me. I spent a hard time trying to understand why the proposed method is robust to noise, until I understand common decision boundary in the very late part of the paper.**
>
> **A3**: We appreciate the reviewer's valuable suggestions. We will carefully revise our paper accordingly.

---

> > ### Comment · Reviewer_BMo4 · 2021-08-22
> > **After rebuttal**
> >
> > I thank the authors for making clarifications. On originality and significance, I still cannot get evidence from either other reviewers or the authors about whether the proposed technique is a straightforward generalization of some existing paper (that generates robust counterfactural explanations for NNs, CNNs, RNNs) to the GNN case here. It seems to me the proposed solution of leveraging common decision boundaries is not specific to GNNs, but generally applicable to other NNs. I hope the AC certify this point before making a decision on this paper. I'd also like to hear from the authors on it and urge the authors to discuss counterfactual explanation methods for general NNs.

---

> > > ### Author Response · Authors · 2021-08-23
> > > **Response to Reviewer BMo4 (Re: After rebuttal)**
> > >
> > > **We thank the reviewer for the reply. Below is our response.**
> > >
> > >
> > > Our method consists of two major steps. 1) Modelling and extracting the decision regions that carry the common decision logic of a GCN; and 2) using the extracted decision regions to produce robust counterfactual explanations on GCNs.
> > >
> > > **To the best of our knowledge, the proposed method is the first to leverage the decision regions of a GCN and produce robust counterfactual explanations on GCNs.**
> > >
> > > To assess the originality and significance of this work, we answer two following questions.
> > >
> > > **First, is there any work that produces robust counterfactual explanations on general deep neural networks?**
> > > We did not find any closely related work that focuses on the robustness of counterfactual explanations for general deep neural networks. All of the existing methods for generating counterfactual explanations on NNs are substantially different from our decision-region-based explanation method.
> > >
> > > **Second, is there any work that produces robust explanations on general deep neural networks?**
> > > Yes, there are some studies that focus on robust explanations, but they do not produce counterfactual explanations and cannot be straightforwardly extended to perform our task on GCNs.
> > > We discuss these works and distinguish our work from them as follows.
> > >
> > > [Chu et al. KDD 2018] proposed a closed form solution to compute exact and consistent explanations for the family of Piecewise Linear Neural Networks (PLNN). The key idea of this work is to extract the original convex polytopes of a small scale PLNN in closed form, and then use these convex polytopes to produce explanations. As stated in Lines 149-171, [Chu et al. KDD 2018] inspires our work to some extent. However, [Chu et al. KDD 2018] is only effective on tiny neural networks with tens of hidden neurons. When the number of hidden neurons increases, the original convex polytopes become too small to cover many input instances. This is exactly why we model large decision regions by solving the proposed SCSC problem, which is substantially different from [Chu et al. KDD 2018].
> > >
> > >
> > > [Wang et al. arXiv 2021] recently proposed the robust explanations by leveraging the geometry of activation regions and aggregating the normal vectors of the local decision boundaries for a target input. This work is similar to our work in terms of exploiting the decision boundaries and the geometry of activation regions. However, it only aggregates the local decision boundaries for a single instance to generate explanations for the given instance. It does not focus on modelling the common decision logic of the model shared by multiple instances to obtain the explanations. This is substantially different from our idea of decision region, which is designed to carry the common decision logic shared by a large number of instances of the same class so that resulting explanations could be more robust and faithful.
> > >
> > > [Alvarez-Melis et al. NeurIPS 2018] studied the stability of explanations and concluded that many previous explanation methods do not produce stable explanations. To ensure similar input data instances get similar explanations, their method directly focuses on designing self-explaining classifier models by enforcing the stability of explanations through regularizations specifically tailored to target neural network models. These stability regularizations are substantially different from our idea of decision region extraction. Additionally, our method doesn’t modify the architecture of the given classifier or retrains it, but interprets the given GNN classifier already trained by the user.
> > >
> > > [Ghorbani et al. AAAI 2019] demonstrated that many classical explanations on image-processing neural networks are not robust to perturbations applied on input data. This work does not propose a method to produce robust explanations, thus it is substantially different from our method.
> > >
> > > [Lakkaraju et al. ICML 2020] proposed a novel framework for generating robust and stable explanations of black box models based on adversarial training. The framework optimizes a minimax objective that aims to construct the highest fidelity explanation with respect to the worst-case over a set of adversarial perturbations. Therefore, it is substantially different from our method.
> > >
> > > [Dombrowski et al. PR 2021] developed a unified theoretical framework for deriving bounds on the maximal manipulability of a model. Based on the theoretical insights, they present three different techniques to boost robustness against manipulation: training with weight decay, smoothing activation functions, and minimizing the Hessian of the network. All these techniques are substantially different from our decision-region-based explanation method.
> > >
> > > In summary, none of the above methods can be straightforwardly extended to produce robust counterfactual explanations on GCNs. In other words, our work here is NOT a straightforward generalization of some existing papers.  Moreover, producing robust counterfactual explanations for general NNs is an interesting direction for future work.  While our study here sheds some light on this direction, our study only focuses on GNNs.  A method for general NNs would have to generalize to tackle challenges with respect to different types of NNs and the corresponding data domains, and may be far from a straightforward extension to our proposed method here.
> > >
> > >
> > >
> > > **References:**
> > >
> > > a)	Chu, Lingyang, Xia Hu, Juhua Hu, Lanjun Wang, and Jian Pei. "Exact and consistent interpretation for piecewise linear neural networks: A closed form solution." In Proceedings of the 24th ACM SIGKDD International Conference on Knowledge Discovery & Data Mining, pp. 1244-1253. 2018.
> > >
> > > b)	Wang, Zifan, Matt Fredrikson, and Anupam Datta. "Boundary Attributions Provide Normal (Vector) Explanations." arXiv preprint arXiv:2103.11257 (2021).
> > >
> > > c)	Alvarez-Melis, David, and Tommi S. Jaakkola. "Towards robust interpretability with self-explaining neural networks." In Proceedings of the 32nd International Conference on Neural Information Processing Systems, pp. 7786-7795. 2018.
> > >
> > > d)	Ghorbani, Amirata, Abubakar Abid, and James Zou. "Explanation of neural networks is fragile." In Proceedings of the AAAI Conference on Artificial Intelligence, vol. 33, no. 01, pp. 3681-3688. 2019.
> > >
> > > e)	Lakkaraju, Himabindu, Nino Arsov, and Osbert Bastani. "Robust and stable black box explanations." In International Conference on Machine Learning, pp. 5628-5638. PMLR, 2020.
> > >
> > > f)	Dombrowski, Ann-Kathrin, Christopher J. Anders, Klaus-Robert Müller, and Pan Kessel. "Towards robust explanations for deep neural networks." Pattern Recognition (2021): 108194.

---

### Official Review · Reviewer_1Uri · 2021-07-19

**Rating:** 7
**Confidence:** 3

**Summary:**

The paper deals with the problem of generating counterfactual explanations for GNNs. The proposed approach is based on directly optimizing the counterfactual explanations in an end-to-end learning scheme. The proposed framework is validated on both synthetic and real data.


**Ethical Concerns:**

No ethical issues occur from this paper.

**Limitations And Societal Impact:**

The authors elaborate on the limitations and societal impact in the conclusion section. The points that they raise are valid. I would add as a limitation of the existing work the lack of ability to deal with weighted edges.

**Main Review:**

Originality: The problem of generating counterfactual explanations form GNNs is very interesting, and worth investigating. The proposed methodology is definitely relevant and valid. Contrary to the existing work, the proposed approach 1) explicitly models the common decision logic of GNNs on similar input graphs and 2) tends to promote that removing the set of edges identified by an explanation from the input
graph changes the prediction significantly.

Quality: The paper is technically sound. The framework is well validated with experimental results on both real and synthetic data. The discussion on the performance of the method on the mutagenicity dataset (in terms of AUC) is interesting. I would expect that a similar discussion on how the graph model/topology affect the performance of the algorithm to be very informative.

Clarity: The paper is generally well written. However, adding a block diagram of the architecture in the main text is needed (Fig. 3 of the appendix). Moreover, some qualitative results on the explanations (similar to table 4 of the appendix) and the difference from the existing works would help the reader to position the paper and appreciate the contribution.

Some additional minor comments:

- line 143: the sentence "Because the feature….” Is not grammatically correct.
- Specify what are the GNNs that are used in the experiments.

Significance: The obtain results are interesting and will probably be used by the research community.






**Time Spent Reviewing:**

4

---

> ### Author Response · Authors · 2021-08-10
> **Response to Reviewer 1Uri**
>
> We thank the reviewer for valuable comments. Below is our response.
>
> **Q1: Quality: The paper is technically sound. The framework is well validated with experimental results on both real and synthetic data. The discussion on the performance of the method on the mutagenicity dataset (in terms of AUC) is interesting. I would expect that a similar discussion on how the graph model/topology affect the performance of the algorithm to be very informative.**
>
> **A1**: For experiments, our explanation network f consists of 2 fully connected layers with a ReLU activation and the hidden dimensionality of 64. This is the same architecture of explanation network as used by PGExplainer [21]. We use this architecture of the network for fair comparison with PGExplainer. We also tried some other versions of architectures by increasing fc layers, and the results remain consistent. We will discuss these results in our revised paper.
>
>
> **Q2: Clarity: The paper is generally well written. However, adding a block diagram of the architecture in the main text is needed (Fig. 3 of the appendix). Moreover, some qualitative results on the explanations (similar to table 4 of the appendix) and the difference from the existing works would help the reader to position the paper and appreciate the contribution.**
>
> **A2**: Thanks for the suggestions, we will carefully revise the paper accordingly.
>
>
> **Q3: Specify what are the GNNs that are used in the experiments.**
>
> **A3**: We use the same architecture as used by GNNExplainer [41] and PGExplainer [21] that consists of 3 graph convolution layers with ReLU activations. We mention this in lines 676-678 of Appendix E.

---

> > ### Comment · Reviewer_1Uri · 2021-08-27
> > **After rebuttal**
> >
> > I thank the authors for responding to my questions. Most of the concerns have been addressed, expect from Q1. By 'graph topology/model', I mean the graph itself (e.g., the generating model, ER, SBM etc), and not the explanation network.

---

> > > ### Author Response · Authors · 2021-08-29
> > > **Response to Reviewer 1Uri (Re: After rebuttal)**
> > >
> > > **We thank the reviewer for the positive comments and valuable suggestions.**
> > >
> > > For fair comparison with the baseline methods, the synthetic datasets we use are widely-adopted datasets that are generated following the same routine as the baseline methods [41, 21, 34, 43, 20]. The experimental results on these synthetic datasets demonstrate the advanced performance of our method. The experimental results on the other two widely-adopted real world datasets (i.e., Mutagenicity [18] and NCI1[35]) further confirm the superior performance of our method.
> > >
> > > We agree with the reviewer that is would be more informative to further discuss how other graph generating models may affect the performance of explanations on GNNs. Therefore, we are in process of conducting these experiments, and we will comprehensively discuss this analysis in our revised version of the paper.

---

### Official Review · Reviewer_gQ2w · 2021-07-20

**Rating:** 6
**Confidence:** 4

**Summary:**

This paper studies the problem of generating explanations for graph classifications. In particular, the paper focuses on the robustness of the explanation. A new explanation generation method is proposed where the key is a new learning objective of the generation model. The objective considers the prototype of each class to eliminate the impact of noise. Extensive experiments on benchmark datasets validate the effectiveness and efficiency of the proposed method.

**Main Review:**

Strong points:
1. This paper studies an important problem about the explanation of graph classification.
2. This paper highlights the robustness of explanations and proposes a new method to enhance the robustness.
3. Extensive experiments are conducted.

Weak points:
1. The paper is not well written with many unclear points. For instance,
- Definition 1 is unclear due to the vague descriptions such as "significantly" and "slight changes".
- Is the counterfactual same as the concept in causal theory? If so, the requirement of counterfactual (the first requirement of the explanation) should be revised. In causal theory, the change of outcome variable (prediction) is not necessary for a counterfactual.

2. "For experiments, our explanation network f consists of 2 fully connected layers with a ReLU activation and the hidden dimension of 64." It would be better to explain the intuition of the network design and explore the structure of the explanation network.

3. "CF-GNNExplainer [20] is only included in the results of node classification, because the source code of CF-GNNExplainer is not available and [20] reports performance on only node classification tasks."
This is a ridiculous excuse for not reporting the performance of an important baseline! (This makes me decrease the rating)

4. The authors argue the importance of aligning well between the generated explanation and human intuition. How does the proposed method achieve this target?

**Time Spent Reviewing:**

3

---

> ### Author Response · Authors · 2021-08-10
> **Response to Reviewer gQ2w**
>
> We thank the reviewer for valuable comments. Below is our response.
>
> **Q1.1: Definition 1 is unclear due to the vague descriptions such as "significantly" and "slight changes".**
>
> **A1.1**: The term "significantly" refers to maximizing the change of prediction scores. This is achieved by minimizing the loss of $\mathcal{L}\_{opp}$ in Equation (10). The term "slight changes" refers to minimizing the changes made on an input graph G. This is achieved by the sparsity constraints of $\mathcal{R}\_{sparse}$ and $\mathcal{R}\_{discrete}$ introduced in lines 288-291 of the paper. We will clarify this in our paper.
>
>
> **Q1.2: Is the counterfactual same as the concept in causal theory? If so, the requirement of counterfactual (the first requirement of the explanation) should be revised. In causal theory, the change of outcome variable (prediction) is not necessary for a counterfactual.**
>
> **A1.2**: We are using the concept of counterfactual explanation given in the 4th paragraph of Section 6.1 of the book "Interpretable Machine Learning: A Guide for Making Black Box Models Explainable" by Christoph Molnar (https://christophm.github.io/interpretable-ml-book/). In this book, "A counterfactual explanation of a prediction describes the smallest change to the feature values that changes the prediction to a predefined output."
>
>
> **Q2: "For experiments, our explanation network f consists of 2 fully connected layers with a ReLU activation and the hidden dimension of 64." It would be better to explain the intuition of the network design and explore the structure of the explanation network.**
>
> **A2**: We use the same architecture of the explanation network as used by PGExplainer [21] for fair comparison. We also tried out some other versions of architectures by increasing fc layers, and the results do not change much.
>
>
> **Q3: "CF-GNNExplainer [20] is only included in the results of node classification, because the source code of CF-GNNExplainer is not available and [20] reports performance on only node classification tasks." This is a ridiculous excuse for not reporting the performance of an important baseline! (This makes me decrease the rating)**
>
> **A3**: We understand the reviewer’s concern and address it as follows.
> Though the code of CF-GNNExplainer has become recently available, it is only meant for node classification tasks and does not support graph classification by default (see Conclusion of [20]).
>
> Since CF-GNNExplainer was originally designed for node classification, graph convolutions were done only on the computation graph of the target node. To extend this to graph classification, graph convolutions need to be done on the entire graph and final embeddings of all nodes should be pooled to predict the class. With these modifications, we have been able to extend CF-GNNExplainer to graph classification and use it to conduct thorough experiments mentioned in Section 5 of the paper. Below are the detailed results.
>
> https://drive.google.com/file/d/10C0wXG9Xsqljzef39ZwTgvgi3ea__-MF/view?usp=sharing
>
> In the following, we analyze the Fidelity, Robustness and Efficiency performance reported in the above result.
>
> **Fidelity**:
> We find that CF-GNNExplainer demonstrates better fidelity than rest of the baselines. This is expected as it is designed to generate counterfactual explanations which are expected to demonstrate higher fidelity. However, it performs significantly worse than our method. This is perhaps because it optimizes each sample independently and does not explicitly use the decision logic of GNN that is shared by group of input graphs.
>
> **Robustness**:
> We also see that its robustness is substantially worse than our method.  This demonstrates how optimizing each sample independently makes it sensitive to artifacts in input graph, and its explanations can change dramatically after applying slight perturbations to the input graph.
>
> **Efficiency**:
> Lastly, its inference runtime is significantly worse than our method and PGExplainer. This is because, like GNNExplainer, it needs to be optimized separately on each input graph and cannot be used in inductive setting.
>
> We will add these results to our paper and make our evaluation code public.
>
>
> **Q4: The authors argue the importance of aligning well between the generated explanation and human intuition. How does the proposed method achieve this target?**
>
> **A4**: "aligning well with human intuition" means a counterfactual explanation is easy to be comprehended by a human. It has been demonstrated by psychologists that counterfactuals elicit causal reasoning in humans [1, 2, 3] and are thus intuitive. There have also been user studies to confirm that users prefer counterfactual explanations over case-based reasoning [4, 5].
>
> Our method is designed to produce counterfactual explanations. Thus, the produced explanations are expected to be easy to comprehend. We have also shown some qualitative results in Table 4 of Appendix H.  RCExplainer can accurately identify the target motifs that aligns well with human understanding. We will clarify this in the revised paper.
>
>
> [1] Ruth Byrne. 2008. The Rational Imagination: How People Create Alternatives to Reality. The Behavioral and brain sciences 30 (12 2008), 439–53; discussion 453. https://doi.org/10.1017/S0140525X07002579
>
> [2] Ruth M. J. Byrne. 2019. Counterfactuals in Explainable Artificial Intelligence (XAI): Evidence from Human Reasoning. In Proceedings of the Twenty-Eighth International Joint Conference on Artificial Intelligence, IJCAI-19. International Joint Conferences on Artificial Intelligence Organization, California, USA, 6276– 6282. https://doi.org/10.24963/ijcai.2019/876
>
> [3] D. Kahneman and D. Miller. 1986. Norm Theory: Comparing Reality to Its Alternatives. Psychological Review 93 (1986), 136–153
>
> [4] Reuben Binns, Max Van Kleek, Michael Veale, Ulrik Lyngs, Jun Zhao, and Nigel Shadbolt. 2018. ’It’s Reducing a Human Being to a Percentage’: Perceptions of Justice in Algorithmic Decisions. In Proceedings of the 2018 CHI Conference on Human Factors in Computing Systems (CHI ’18). Association for Computing Machinery, New York, NY, USA, 1–14. https://doi.org/10.1145/3173574.3173951
>
> [5] Jonathan Dodge, Q. Vera Liao, Yunfeng Zhang, Rachel K. E. Bellamy, and Casey Dugan. 2019. Explaining Models: An Empirical Study of How Explanations Impact Fairness Judgment. In Proceedings of the 24th International Conference on Intelligent User Interfaces (IUI ’19). Association for Computing Machinery, New York, NY, USA, 275–285. https://doi.org/10.1145/3301275.3302310

---

> > ### Comment · Reviewer_gQ2w · 2021-08-24
> > **After Rebuttal**
> >
> > Thanks to the authors for the explanations.
> >
> > Most of my concerns are addressed except Q4. After looking through the papers listed, it is hard for me to find out strong connections between human intuition and the explanations generated by the proposed method. As mentioned by [1], the rational imaginations of us humans are related to strong and weak causal relations. It is unclear whether the method indeed considers such relations. As such, I suggest omitting the argument in the revision.

---

> > > ### Author Response · Authors · 2021-08-24
> > > **Response to Reviewer gQ2w (Re: After Rebuttal)**
> > >
> > > We thank the reviewer for the reply . We will address the reviewer’s remaining concern by following the reviewer’s suggestion and removing the claim that our explanations “align well with the human intuition”.

---

### Official Review · Reviewer_Vbh5 · 2021-07-23

**Rating:** 5
**Confidence:** 3

**Summary:**

This paper introduces a robustness explanations method on graph neural networks (RCExplainer) against random noise. To achieve their goals, they first model the common decision logic of a GNN and then extract robust counterfactual explanations by neural networks that explore the decision logic carried by the linear decision boundaries of the decision regions.
Their conducted experiments have verified the effectiveness of the proposed framework.

**Ethical Concerns:**

N.A

**Ethics Review Area:**

["I don’t know"]

**Limitations And Societal Impact:**

N.A

**Main Review:**

There are several major concerns in this work.

1. Authors may put more effort to improve the presentation/writing of the paper.


2. The task is a little bit weird for me.
 Can random noise affect/change the GNN's prediction (e.g., node's label)?
If yes,  it would be better to investigate whether there are some correlations between GNNs and explainer.
It's not easy to understand how to conduct a robust GNN Explainer on non-robust GNNs. If the perturbations can change the GNN's prediction, it has a high chance to change the subgraph generated by the proposed method.

Although this work doesn't work on adversarial attacks on GNNs model, they are suggested to perturb the graph data via some GNNs attacking methods to solid their methods, instead of random noise.

How to add the random noise? Is it for the entire graph data or individual node?
For the explanation on a target node,  if added noise is far away from the local neighbors of the target node, it's meaningless to study their robustness and it's also hard to change the explanation results.

3. Modeling decision regions for robust explainer is not well motivated.
Are there any other solutions? What are the detailed advantages of modeling decision boundaries for train GNN Explainers in their proposed methods?

4. It would be better to analyze the computation time for training, although the proposed method doesn't require too much time for inference.
The inference time of their methods basically depends on their designed neural networks. As they only compute a score on the embedding of two nodes, it works very efficiently.

5. CF-GNNExplainer ->"CF-GNNExplainer [20] is only included in the results of node classification, because the source code of CF-GNNExplainer is not available "
The code has already been released three months ago. You can check at GitHub.

**Time Spent Reviewing:**

3

---

> ### Author Response · Authors · 2021-08-10
> **Response to Reviewer Vbh5 - part 2/2**
>
> **Q5: CF-GNNExplainer ->"CF-GNNExplainer [20] is only included in the results of node classification, because the source code of CF-GNNExplainer is not available " The code has already been released three months ago. You can check at GitHub.**
>
> **A5**: We thank the reviewer for pointing to the code base of CF-GNNExplainer. Though the code of CF-GNNExplainer has become recently available, it is only meant for node classification tasks and does not support graph classification by default (see Conclusion of [20]).
>
> Since CF-GNNExplainer was originally designed for node classification, graph convolutions were done only on the computation graph of the target node. To extend this to graph classification, graph convolutions need to be done on the entire graph and final embeddings of all nodes should be pooled to predict the class. With these modifications, we have been able to extend CF-GNNExplainer to graph classification and use it to conduct thorough experiments mentioned in Section 5 of the paper. Below are the detailed results.
>
> https://drive.google.com/file/d/10C0wXG9Xsqljzef39ZwTgvgi3ea__-MF/view?usp=sharing
>
> In the following, we analyze the Fidelity, Robustness and Efficiency performance reported in the above result.
>
> **Fidelity**:
> We find that CF-GNNExplainer demonstrates better fidelity than rest of the baselines. This is expected as it is designed to generate counterfactual explanations which are expected to demonstrate higher fidelity. However, it performs significantly worse than our method. This is perhaps because it optimizes each sample independently and does not explicitly use the decision logic of GNN that is shared by group of input graphs.
>
> **Robustness**:
> We also see that its robustness is substantially worse than our method.  This demonstrates how optimizing each sample independently makes it sensitive to artifacts in input graph, and its explanations can change dramatically after applying slight perturbations to the input graph.
>
> **Efficiency**:
> Lastly, its inference runtime is significantly worse than our method and PGExplainer. This is because, like GNNExplainer, it needs to be optimized separately on each input graph and cannot be used in inductive setting.
>
> We will add these results to our paper and make our evaluation code public.

---

> ### Author Response · Authors · 2021-08-10
> **Response to Reviewer Vbh5 - part 1/2**
>
> We thank the reviewer for valuable comments. Below is our response.
>
> **Q1: Authors may put more effort to improve the presentation/writing of the paper.**
>
> **A1**: We will make our best effort to further improve the paper.
>
>
> **Q2.1: The task is a little bit weird for me. Can random noise affect/change the GNN's prediction (e.g., node's label)? If yes, it would be better to investigate whether there are some correlations between GNNs and explainer. It's not easy to understand how to conduct a robust GNN Explainer on non-robust GNNs. If the perturbations can change the GNN's prediction, it has a high chance to change the subgraph generated by the proposed method.**
>
> **A2.1**: Robustness of GNN is an important direction and significant progress has been made in this domain [1,2].  However, our paper tackles an essentially different task: given a GNN that may or may not be robust, how can we obtain robust explanation against random noise.  We elaborate the robustness of explanations as follows.
>
> Given an input graph that is perturbed/modified by random noise, if the GNN's prediction on the perturbed input graph does not change, then the corresponding explanation should not change dramatically. Following this intuition, if the GNN’s prediction on the perturbed input graph does not change and the explanation does not change dramatically, then we say the explanation is “robust”. Otherwise, if the explanation changes dramatically when the GNN’s prediction is not changed, then we say the explanation is not robust.
>
> Therefore, no matter whether a GNN is robust or not to random noise, it is still meaningful to evaluate the robustness of an explanation to random noise when a GNN’s prediction is not changed by random noise. Following this idea, when evaluating the robustness performance shown in Figure 2, we only consider the perturbed input graphs whose predictions are not changed by random noise. We mention this in lines 349-352 of the paper.
>
> We will conduct some additional experiments on both robust and non-robust GNN models separately to clarify this point further in the revised paper.
>
>
> [1] Wei Jin, Yao Ma, Xiaorui Liu, Xianfeng Tang, Suhang Wang, and Jiliang Tang. 2020. Graph Structure Learning for Robust Graph Neural Networks. Proceedings of the 26th ACM SIGKDD International Conference on Knowledge Discovery & Data Mining. Association for Computing Machinery, New York, NY, USA, 66–74. DOI:https://doi.org/10.1145/3394486.3403049
>
> [2] Geisler, S., Zügner, D., & Günnemann, S. (2020). Reliable graph neural networks via robust aggregation, NeurIPS’20,. arXiv preprint arXiv:2010.15651.
>
> **Q2.2: Although this work doesn’t work on adversarial attacks on GNNs model, they are suggested to perturb the graph data via some GNNs attacking methods to solid their methods, instead of random noise.**
>
> **A2.2**: When a GNN model is attacked (successfully) by an adversarial attack, the model itself is not robust against the attack. As the reviewer pointed out in Q2.1, it is meaningless to discuss robust explanation in this situation where the prediction of GNN changes. However, if the adversarial attack on GNN fails and the prediction remains consistent, then the evaluating robustness of explanation would be meaningful. We will add this setting to our experiments in the revised paper.
>
>
> **Q2.3: How to add the random noise? Is it for the entire graph data or individual node? For the explanation on a target node, if added noise is far away from the local neighbors of the target node, it's meaningless to study their robustness and it's also hard to change the explanation results.**
>
> **A2.3**: We evaluate the explanation performance on two tasks: node classification task and graph classification task. The ways to add noise are slightly different in these two tasks.
>
> For node classification task, as mentioned in lines 578-594 of Appendix B, the prediction of GNN on a target node is determined by the computation graph of node. The computation graph of a node is defined as the k-hop neighborhood of the target node where k is the number of convolutions used in the GNN. Applying noise to the areas other than computational graph of the target node is not a meaningful way to study the robustness of explanations. Hence, noise should only be applied uniformly to the computation graph of the target node, as suggested in your comment. For graph classification task, as illustrated in lines 350-352, we add noise uniformly to the complete graph because the prediction of GNN is dependent upon the features of whole graph. We will clarify the above discussion further in the paper.
>
> **Q3: Modeling decision regions for robust explainer is not well motivated. Are there any other solutions? What are the detailed advantages of modeling decision boundaries for train GNN Explainers in their proposed methods?**
>
> **A3**: We motivate modelling decision regions for robust explainer in several places of the paper. In lines 105-108, we introduce the intuition that a GNN’s common decision logic on a large group of similar input graphs do not easily overfit the noise of an individual input graph, therefore, explanations produced from the common decision logic of a GNN will be more robust to noise. In lines 135-137, we introduced that the key idea to model a GNN’s common decision logic is to find the decision regions of the GNN. In lines 159-171, we conclude why modelling the common decision logic of a GNN by decision regions that cover many similar input graphs can produce robust explanations. In summary, using decision regions to model the common decision logic of a GNN is the key to produce robust explanations, because a GNN’s common decision logic is shared by a large group of similar input graphs, thus explanations generated from common decision logic do not easily overfit the noise of an individual input graph.
>
> To the best of our knowledge, there are no other GNN explainer methods that explicitly model the decision boundaries of GNN to derive the explanations for GNN predictions. Our work is novel in two major ways: 1) our method is the first to focus on generating robust counterfactual explanations for GNN predictions; 2) we are the first ones to leverage common decision boundaries of the GNN model to generate robust explanations for GNN’s predictions.
>
> As discussed in the related work section and demonstrated by our experiments in Figure 2, explanations produced by most existing methods are not robust to random noise. This is because they independently optimize the correlation between a single input graph and the corresponding prediction made by a GNN, but do not explicitly leverage the real decision logic of GNNs that is shared by a group of input graphs. As a result, some noisy nodes and edges may achieve a very high correlation with the prediction score that might cause these methods to miss patterns that generally exist in many similar input graphs and are the right explanations of GNN’s predictions. Due to this sensitivity to noise, even small modifications when applied to an input graph that do not change the GNN’s prediction may cause the explanation to change dramatically.
>
> Now we discuss the advantage of modeling decision boundaries and decision regions of a trained GNN. The decision regions are modelled by convex polytopes that consist of the decision boundaries of the GNN. As illustrated in lines 149-158, according to [4], the convex polytopes formed by the decision boundaries of a GNN truthfully carry the decision logic of the GNN. The advantage of forming convex polytopes by the decision boundaries of GNN is that explanations produced this way will be truthful to the decision boundary of the GNN, thus they will reflect the real decision logic of the GNN. Moreover, since the decision boundaries of a convex polytope carry the common decision logic shared by a large number of input graphs contained in the convex polytope, explanations generated from the common decision logic will be highly robust to noise.
>
>
>
> **Q4: It would be better to analyze the computation time for training, although the proposed method doesn't require too much time for inference. The inference time of their methods basically depends on their designed neural networks. As they only compute a score on the embedding of two nodes, it works very efficiently.**
>
> **A4**: Both PGExplainer and RCExplainer can be trained and then be used in inference mode on new unseen input graphs. Here is the training time comparison of ours vs PGExplainer on different datasets:
>
>
>
> ||Mutagenicity|BA_2Motifs|NCI1|
> |--|----------|----------|----|
> |PGExplainer| 30.0s ± 3.5|5.5s ± 1.3|30.7s ± 4.5|
> |RCExplainer (Ours)|33.2s ± 2.4|5.8s ± 1.4|29.2s ± 3.6|
>
>
>
> The mentioned training time (in seconds) is per epoch and we use 500 epochs in training of both the methods. As shown, the training time of our method (i.e., RCExplainer) is comparable with training time of PGExplainer. The specifications of the system and GPU used for training are mentioned in lines 676-688 of Appendix E.
>
> We also found that using Pytorch Geometric implementations of GNN significantly speeds up the training for the above methods by a factor of 2x-5x. Training is not applicable to other baselines as they are directly used to optimize over the given input graph. Therefore, their training time is not available in the above table.
>
> **Q5 is answered in the next comment.**

---

> > ### Comment · Reviewer_Vbh5 · 2021-08-24
> > **After Rebuttal**
> >
> > Thanks to the authors' response to my concerns. Most of my concerns are well addressed except Q2.
> >
> > I agree that, given an input graph that is perturbed/modified by random noise/adversarial attacks,
> >
> > (a) GNN's prediction does not change, the corresponding explanation should not change dramatically. --> Robust Explainer.
> >
> > (b) If the GNN’s prediction is not changed, the explanation changes dramatically. --> not Robust Explainer
> >
> > Also,  it is very important and meaningful to investigate the robustness in such a situation: if a GNN model is successfully attacked, the explanations would change or not. If not, it's then not a robust explainer. Some works have already studies that explanations of the deep models can be attacked as well [1,2, 3].
> >
> >
> > [1] Explanation of neural networks is fragile
> >
> > [2] Proper Network Interpretability Helps Adversarial Robustness in Classification
> >
> > [3] Jointly Attacking Graph Neural Network and its Explanations

---

> > > ### Author Response · Authors · 2021-08-28
> > > **Response to Reviewer Vbh5 (Re: After Rebuttal)**
> > >
> > > **We thank the reviewer for the valuable comments. We address the reviewer's remaining concern as follows.**
> > >
> > >
> > >
> > > **Q6: Also, it is very important and meaningful to investigate the robustness in such a situation: if a GNN model is successfully attacked, the explanations would change or not. If not, it's then not a robust explainer.**
> > >
> > > **A6:** In the references provided by the reviewer, there are **two different notions of the robustness of explanations**.
> > >
> > > The **first notion** is the robustness proposed in [Ghorbani et al. AAAI 2019]. As concluded by the reviewer, the key idea of this notion is:
> > > **a)**	If a GNN's prediction does not change, the corresponding explanation should not change dramatically. => Robust Explainer.
> > > **b)**	If the GNN’s prediction is not changed, the explanation changes dramatically. => not Robust Explainer
> > >
> > > The **second notion** aligns with the concepts proposed in [Boopathy et al. ICML 2020] and [Fan et al. arXiv 2021]. This notion can be concluded as:
> > > **c)**	If a GNN’s prediction is changed by a perturbation on an input graph, but the explanation does not change. => not Robust Explainer
> > > **d)**	If GNN’s prediction is changed by a perturbation on an input graph, and the explanation changes. => Robust Explainer.
> > >
> > > **The two notions correspond to two substantially different perspectives of the robustness of explanations.** Both notions are important and meaningful. Our work focuses on producing robust explanations based on the first notion of robustness. While [Boopathy et al. ICML 2020] focuses on producing robust explanations for CNNs based on the second notion of robustness, and [Fan et al. arXiv 2021] focuses on attacking GNN explanation methods based on the second notion of explanation robustness. Therefore, the focus of our work is substantially different from [Boopathy et al. ICML 2020] and [Fan et al. arXiv 2021].
> > >
> > > We agree with the reviewer that the second notion of robustness is equally important and meaningful to investigate. Therefore, we conducted the following **additional experiments** to further study the performance of our method based on the second notion of robustness.
> > >
> > > Denote by G an input graph and by G’ the perturbed input graph whose label is successfully changed by perturbing G. Denote by S and S’ the explanations on G and G’, respectively. We measure the change of explanations by the AUC of ROC between S and S’ as illustrated in Lines 353-355 of the paper. A higher AUC means a smaller difference between S and S’; **a lower AUC means a larger difference between S and S’.**
> > > Based on the second notion of robustness, since the label of G’ is changed by perturbation, a robust explanation S’ should be different from S. **Therefore, a lower AUC implies a better explanation robustness.**
> > >
> > > The new experiments are conducted on three graph classification datasets: Mutagenicity, BA-2motifs and NCI1, in a similar setting as the robustness experiments in Section 5.2. We list the only **two differences in settings** as follows.
> > >
> > > **1>**	In the new experiments, **the prediction of GNN on G’ is successfully changed by perturbation** applied on G. However, in the existing experiment in Section 5.2, the prediction on G’ was not changed by perturbation.
> > > **2>**	In the new experiments, we applied two versions of perturbations including:  **i) random noise** that is the same noise we used in Section 5.2; and **ii) adversarial noise** generated by the source code of RL-S2V published by [Dai et al. ICML 2018].
> > >
> > >
> > > **The experimental results are attached as follows.**
> > >
> > > https://drive.google.com/file/d/1V8Y2Q0jsyx_I9HftG972qzqSHSJlDMoD/view?usp=sharing
> > >
> > >
> > > **Figure 1** shows the results when random noise is applied to perturb input graphs. Our method achieves a substantially lower AUC than the other baseline methods on Mutagenicity and NCI1 datasets. On BA-2motifs, our method again has the best performance except at some higher levels of noise where PGExplainer has slightly better performance than RCExplainer. This indicates that our method is **more robust to random noise than the other baseline methods** based on the second notion of explanation robustness.
> > >
> > > **Figure 2** shows the results when adversarial noise generated by the source code of [Dai et al. ICML 2018] is applied to perturb the input graphs.  We observe similar results as before, as our method again achieves a much lower AUC than the other baseline methods. This further indicates that our method is **more robust to adversarial noise than the other baseline methods** based on the second notion of explanation robustness.
> > >
> > > **In conclusion, even based on the second notion of explanation robustness, our method is more robust than the other baseline explanation methods. We will comprehensively discuss the above in the revised version of the paper.**
> > >
> > > **References**
> > >
> > > i.	Ghorbani, Amirata, Abubakar Abid, and James Zou. "Interpretation of neural networks is fragile." In Proceedings of the AAAI Conference on Artificial Intelligence, vol. 33, no. 01, pp. 3681-3688. 2019.
> > >
> > > ii.	Boopathy, Akhilan, Sijia Liu, Gaoyuan Zhang, Cynthia Liu, Pin-Yu Chen, Shiyu Chang, and Luca Daniel. "Proper network interpretability helps adversarial robustness in classification." In International Conference on Machine Learning, pp. 1014-1023. PMLR, 2020.
> > >
> > > iii.	Fan, Wenqi, Wei Jin, Xiaorui Liu, Han Xu, Xianfeng Tang, Suhang Wang, Qing Li, Jiliang Tang, Jianping Wang, and Charu Aggarwal. "Jointly Attacking Graph Neural Network and its Explanations." arXiv preprint arXiv:2108.03388 (2021).
> > >
> > > iv.	Dai H, Li H, Tian T, Huang X, Wang L, Zhu J, Song L. Adversarial attack on graph structured data. In International Conference on Machine Learning 2018 Jul 3 (pp. 1115-1124). PMLR.

---

### Public Comment · ~Zhaoning_Yu2 · 2021-11-28
**Could you share your code through github?**

Hi authors,

I am really interested in your paper.
But I could not find the code through the link you provided in the paper.
Could you share the code through Github?

The notebook on huaweicloud is empty.

Thank you so much,
Zhaoning

---

> ### Public Comment · ~Lingyang_Chu1 · 2021-11-28
> **Code will be ready soon.**
>
> Hi Zhaoning Yu,
>
> Thank you very much for your interest in our work.
>
> The code will be available on huaweicloud in a couple of weeks.
>
> Best,

---

> ### Public Comment · ~Lingyang_Chu1 · 2021-12-04
> **Code available now.**
>
> Hi Zhaoning Yu,
>
> The code of this paper is now available at the URL provided on openreview.
>
> Here is a copy of the URL.
>
> https://developer.huaweicloud.com/develop/aigallery/notebook/detail?id=e41f63d3-e346-4891-bf6a-40e64b4a3278
>
> Thanks for your interest and patience.
>
> Best,
>
> Lingyang Chu

---

### Public Comment · ~Amir_Reza_Mohammadi1 · 2023-11-06
**Shared code link is not Working!!!**

Dear Authors,
Your shared link for the code is not working. As already mentioned by Zhaoning Yu, why not sharing it on github like others?

thanks in advance

(Jan 2024: The shared linked on Nov. 6th. 2023 is now functional. Thanks dear Lingyang.)

---

### Public Comment · ~Lingyang_Chu1 · 2023-11-06
**New URL for the code (updated on Nov. 6th, 2023)**

Please use this URL (updated on Nov. 6th, 2023): https://developer.huaweicloud.com/develop/aigallery/notebook/detail?id=e41f63d3-e346-4891-bf6a-40e64b4a3278

We could not put it on Github due to internal reasons.

---

### Decision · Program_Chairs · 2021-09-28

**Decision:**

Accept (Poster)

**Comment:**

Overall the reviewers agree that this work contributes a valid, useful method for robust counterfactual explanations on GNNs (itself of significant importance) that is well evaluated and explained. Note the reviewers have pointed out several places where the write up could be improved, and the authors are advised to address these in the final version of the work.

**Consistency Experiment:**

NeurIPS has a long history of experimentation. In 2014, NeurIPS ran an experiment in which 10% of submissions were reviewed by two independent committees to quantify the randomness in the review process. This year, we repeated a variant of this experiment to see how the quality of the review process has changed over time.  This paper was part of the experiment and was therefore assigned to two committees (consisting of reviewers, an Area Chair, and a Senior Area Chair) that reached independent decisions.  If both committees made the same recommendation, this recommendation was followed. If a single committee recommended acceptance, the paper was accepted (with the exception of a few cases in which the other committee identified what we considered a fatal flaw, e.g., an error in a key result).

This copy’s committee reached the following decision: **Accept (Poster)**

The other committee assigned to the paper recommended **Reject**.  You can find the other set of reviews, along with any follow up discussion with the authors here:
https://openreview.net/forum?id=wGmOLwb8ClT